# Photo-Oxidation of Therapeutic Protein Formulations: From Radical Formation to Analytical Techniques

**DOI:** 10.3390/pharmaceutics14010072

**Published:** 2021-12-28

**Authors:** Elena Hipper, Michaela Blech, Dariush Hinderberger, Patrick Garidel, Wolfgang Kaiser

**Affiliations:** 1Institute of Chemistry, Martin-Luther-Universität Halle-Wittenberg, von-Danckelmann-Platz 4, 06120 Halle (Saale), Germany; elena.hipper.ext@boehringer-ingelheim.com (E.H.); dariush.hinderberger@chemie.uni-halle.de (D.H.); 2Boehringer Ingelheim Pharma GmbH & Co. KG, Innovation Unit, PDB, Birkendorfer Strasse 65, 88397 Biberach an der Riss, Germany; michaela.blech@boehringer-ingelheim.com

**Keywords:** photo-degradation, analytical methods, post-translational modification, protein degradation, reactive oxygen species

## Abstract

UV and ambient light-induced modifications and related degradation of therapeutic proteins are observed during manufacturing and storage. Therefore, to ensure product quality, protein formulations need to be analyzed with respect to photo-degradation processes and eventually protected from light exposure. This task usually demands the application and combination of various analytical methods. This review addresses analytical aspects of investigating photo-oxidation products and related mediators such as reactive oxygen species generated via UV and ambient light with well-established and novel techniques.

## 1. Introduction

Therapeutic proteins such as monoclonal antibodies (mAbs) are important alternatives to small chemical drugs. The benefits of therapeutic proteins are their high activity and specificity in combination with a reduced risk for side effects [1]. Unfortunately, due to their complex structure, therapeutic proteins are quite susceptible to various environmental parameters such as temperature or light [2,3,4], which especially impede manufacturing and storage [5]. It is described in the literature that the exposure of therapeutic proteins to light encountered during, e.g., common manufacturing processes such as purification, fill and finish, storage or administration in the clinic, may promote the formation of various protein modifications [6,7,8,9], which might show immunological effects [10]. These aberrant byproducts can be generated via two photo-oxidation processes that proceed either via a so-called type I or type II oxidation pathway [11]. In essence, both pathways are initiated via the excitation of a chromophore, in this context referred to as photo-sensitizer (sen) by the absorption of light as depicted in Figure 1. The subsequent conversion of the excited photo-sensitizer singlet state (^1^sen) to its triplet state (^3^sen) via intersystem crossing (ISC) is a prerequisite for photo-oxidation processes. In its triplet state, the sensitizer returns to its ground state by transferring the previously absorbed light energy via an electron (type I) or energy (type II) transfer mechanism. Consequently, the type I pathway results in the formation of radicals and radical ions. These species often lead to subsequent modification of the substrate molecule and the sensitizer itself. Furthermore, due to the transfer of electrons and the radical character of the reaction, consecutive radical chain reactions are typical outcomes of the type I pathway generating a broad variety of photo-oxidation products. The formation of singlet oxygen (^1^O_2_) via energy transfer from the ^3^sen to an oxygen molecule is the characteristic trait of the type II pathway. Hence, formation of chemical modifications [9,12], such as protein peroxides [13,14], crosslinks [15,16], fragmentation [17,18] or reactive oxygen species (ROS) formation [11] have been observed due to photo-oxidation processes. However, the generation of various degradation products depends on the light exposure [19] and conditions in place and its corresponding parameters, such as light intensity or wavelength distribution, which often leads to distinct photo-oxidation patterns [20]. In this regard, realistic light conditions for therapeutic proteins are ranging from 1000 lux in laboratories [21] to 130,000 lux under direct sunlight [22], and the most dominant spectral regions are the UV-A (315–400 nm) and visible light (ca. 400–780 nm). Hence, most photo-oxidation processes observed in therapeutic protein formulations are not based on proteinogenic sensitizer such as Trp or Tyr, since these solely absorb light in the UV-C (200–280 nm, FUV) and UV-B (280–315 nm) regions. Impurities from cell culture such as B vitamins [23] might introduce photooxidation processes. Worth mentioning, wavelengths in the UV-A and UV-B region are usually absorbed by commercially available window glass or medicinal containers such as glass vials or polyethylene infusion bags [24]. Consequently, common light conditions do not directly damage therapeutic proteins. Most protein modifications are the results of non-proteinogenic photo-sensitizers, which are producing ROS. Hence, ROS are key components of photo-oxidation processes that target therapeutic proteins, excipients, and surfactant molecules alike. Therefore, their emergence may be a concern for the stability and efficacy of therapeutic protein formulations.

The aim of the present review is to provide an updated and broad overview of analytical techniques for investigating photo-induced alterations of therapeutic protein formulation components with the main focus on ROS formation via photo-chemical reactions and resulting protein modifications.

## 2. Measurement of Light Intensity

The ICH guideline on photostability (ICH Topic Q1B) describes a useful basic protocol for testing new drug substances for manufacturing, storage and distribution processes. The guideline describes the illumination conditions such as light source and overall illumination and exposure time. Two different light studies were distinguished, a forced degradation study and a confirmatory study. The forced degradation study should characterize the intrinsic stabilities of the drug substance or drug product, while the confirmation study should depict data of ambient light conditions and whether precautionary measures are needed.

The basis of light investigation is the measurement of light intensity, depending on the wavelengths of the light. Two measurement principles are distinguished: On the one hand, radiometers, devices measuring the radiant flux (power) of electromagnetic radiation with sensitivity to defined wavelength ranges. The amount of incoming radiation is expressed by the number of photons of a certain wavelength passing through a unit in a specific time [4]. In radiometry, the radiation flux includes all electromagnetic waves in terms of absolute power. In photometry, the radiant power at each wavelength is weighted by a luminosity function that models human brightness sensitivity. The irradiance is called illuminance which is measured in lux (power per unit area; mW∙m^−2^) and lumen (lm) is the unit of power (in Watt; W). Lumen as the photometric International Systems of Units (SI) derived unit of luminous flux, is a measure of the total quantity of visible light emitted by a source per unit of time. It takes into account the sensitivity of the human eye i.e., the luminous flux differs from power (radiant flux) in that radiant flux includes all electromagnetic waves emitted, while luminous flux is weighted according to a model (a “luminosity function”) of the human eye’s sensitivity to various wavelengths [25]. One lumen is defined as the luminous flux of a 1.464 mW at a defined wavelength of 555 nm with 100% efficiency. The light intensity has to be determined with different instruments according to its electromagnetic spectrum. These instruments all have their specific wavelength range, in which these devices are calibrated. In photometry (λ = 400–800 nm) a lux meter is used as a simple radiometer [4]. Lux meters are only capable to measure the luminous flux per unit area of light, expressed in lux units, with 1 lux being equal to 1 lumen per square meter. Since incoming light from different sources differ in their spectrum, the luminous flux cannot be compared unless the lux meter is calibrated for each source [4,26]. Measuring the incoming light through containers, such as primary packaging, is possible with a spectroradiometer [27], which consists of a monochromator with a photomultiplier. Since a spectroradiometer measures the detailed spectral power distribution, the exact intensity of photons can be obtained by the function of wavelengths. A thermopile measures the total radiant flux in UV, visible and IR spectrum as thermal response. It is used to calibrate other instruments. For daily use the thermopile is not practicable, as it is costly and difficult to handle [4]. The measurement of incoming luminous flux can be measured with a lux meter, spectroradiometer and thermopile. These devices all contain the same limitation in that irradiance is measured instead of the absorbed energy. The amount of radiation is reduced when it is transmitted, scattered, and reflected. The absorbed photons can be measured by using chemical actinometers which can be placed in exactly the same optical position as a solution sample.

Chemical actinometers are used for measuring direct radiation, i.e., the number of photons in a beam integrally or per unit time. The measuring principle is based on measuring radiant flux (radiant power) via the yield from a chemical reaction, induced by a chemical with a known quantum yield. Actinometers therefore use chemical compounds, such as potassium ferrioxalate with a sensitive wavelength range λ = 254 to 500 nm or meso-diphenylhelianthrene with a sensitive wavelength range λ = 400 to 700 nm. Changes of these chemical compounds are used to measure the intensity of the radiation during irradiation [28]. An actinometer maps the full range of wavelengths to which the various samples are tested. Different chemical actinometers can be used to cover the electromagnetic spectrum of light [29]. The most commonly used actinometer is the potassium ferrioxalate actinometer. It is also known as Hatchard-Parker actinometer, which relies on the decomposition of tris(oxalato)ferrate(III) (ferrioxalate) ions into Fe^2+^ [30]. The Fe^2+^ ions may be quantified by the complexation with 1,10-phenanthroline [30]. The ferrioxalate actinometer has a high quantum yield of ca. 1.38 ± 0.03 (exactly at 253.7 nm) [31] and therefore covers a wide range in the UV and parts of the visible region (approx. λ = 250–500 nm) [30,32]. A limitation of using the potassium ferrioxalate actinometer is the short reaction time (5 to 10 min). It can therefore not be used in long-term light stability studies [4]. The uranyl oxylate actinometer is mentioned in ICH Guideline Q1B. This actinometer system covers a wide range in the UV-A and visible spectrum with a quantum yield of ca. 1.02 ± 0.01 (λ at 254 nm) [33]. The reaction of the uranyl oxylate actinometer is slower than the reaction of the ferrioxalate actinometer, which allows a longer time period of possible photon absorption. However, the reagents of the uranyl oxylate actinometer are expensive and toxic [4]. For UV measurements, the ICH guideline proposes a quinine actinometer, which can be used in the range between 300–400 nm [34]. The sensitivity of the quinine actinometer depends on experimental conditions such as measurement time interval, pH, temperature [35], oxygenation and the lamp emission spectrum [34,36].

## 3. Photo-Oxidation and Oxygen

Molecular oxygen is a major component in the type I and II photo-oxidation pathways as previously described and depicted in Figure 1. Consequently, both pathways are capable to promote the formation of several ROS with varying oxidation potentials (Figure 2).

### 3.1. Singlet Oxygen (^1^O_2_)

*Singlet oxygen* (^1^O_2_) (Table 1) is generated via the type II pathway, when an excited photo-sensitizer, like rose bengal [39] or riboflavin [40], transfers energy to an oxygen molecule. Thereby, the oxygen molecule is lifted from its triplet ground state (^3^O_2_) to its excited singlet state (^1^O_2_) with both electrons in the antibonding orbitals paired. Compared to ^3^O_2_, singlet oxygen ^1^O_2_ has an oxidation potential of 0.65 V [37] (Figure 2) and a long lifetime of a few microseconds [41]. Therefore, ^1^O_2_ is highly reactive towards organic molecules with pi-electrons or free electron pairs of low ionization energy. Hence, especially amino acid residues such as Trp, Tyr, His, Met, and Cys are readily oxidized by ^1^O_2_ in solution with average reaction rates between 0.2–5·10^7^ M^−1^ s^−1^ [42]. Detection of ^1^O_2_ in solution is frequently achieved by monitoring the characteristic ^1^O_2_ phosphorescence emission band at 1270–1275 nm [43,44,45], that can be observed if the excited singlet oxygen returns to its original triplet state with an average quantum yield of approximately 10^−6^ [43]. The low quantum yield of ca. 10^−6^ and the lack of sensitivity of common IR (infrared) detectors make investigations of ^1^O_2_ especially in aqueous solutions challenging, since the ^1^O_2_ emission signal is further quenched by water molecules [46,47]. Hence, in most studies’ molecular singlet oxygen probes, whose properties are altered by ^1^O_2_, are employed to investigate ^1^O_2_ related phenomena. The reactions with ^1^O_2_ often lead to changes in the electronic structures of chromophores that allows the indirect detection of ^1^O_2_ and other ROS especially via spectroscopic techniques. The probes react with ^1^O_2_ which either significantly enhance or deteriorate their spectroscopic properties like the quantum yield. Hence, the formation of ROS, that cannot be detected directly due to very low yields or the presence of interfering compounds, can still be investigated via a broad variety of molecular probes. In the context of ^1^O_2_, any π-conjugated dienes can serve as absorption-based probes for ^1^O_2_ such as 5-membered heteroarenes, acenes, and alkenes. Regarding regular laboratory equipment and feasibility, fluorescence probes are often used for detecting ^1^O_2_ in solution. Although, various molecular probes seem to be available for investigating ^1^O_2_ in biological samples, most of these probes are poorly soluble in aqueous solutions and/or react readily with other ROS, and thus possess insufficient specificity towards ^1^O_2_ like 1,3-diphenylisobenzofuran (DPBF) [48,49]. In contrast, anthracene based fluorescence probes such as the 9-[2-(3-carboxy-9,10-diphenyl)anthryl]-6-hydroxy-3H-xanthen-3-one (DPAX) [50] are widely used for measuring ^1^O_2_ in biological systems. The anthracene moiety reacts specifically with ^1^O_2_ to form a thermostable endoperoxide (EP) [51], resulting in a significant increase in quantum yield (φ_F_ 0.2–0.7) of the fluorescein moiety [50]. Although a broad variety of fluorescence probes exists, the selection of a suitable molecular probe heavily depends on the experimental conditions and the related issue in question. For instance, the commercially available Singlet Oxygen Sensor Green (SOSG©) [52] or the near-infrared (NIR) probe indocyanine green (ICG) [53,54] are known ^1^O_2_ photo-sensitizers. Consequently, it is not recommended to use these or similar probes for studying photo-oxidation processes. Furthermore, the high sensitivity of fluorescence-based techniques is not always an advantage, especially if the fluorescence signal heavily depends on solvent conditions [55] or is quenched by other molecules present in the solution. However, the modification of anthracene-based probes by ^1^O_2_ can be monitored by UV-Vis spectroscopy. Although this approach is less sensitive compared to fluorescence measurements, it is less prone to artefacts such as solvent conditions or other present quenching molecules. In this context, DPBF or 9,10-anthracenediyl-bis(methylene)dimalonic acid (ABDA) are usually the common probes of choice, whereas the latter shows a much higher degree of specificity towards ^1^O_2_, since ABDA reacts primarily with ^1^O_2_ [56] and only in special circumstances with superoxide anions [57]. Another alternative to fluorescence or UV-Vis probes are lanthanide phosphorescence probes, like [4′-(9-anthryl)-2,2′:6′,2″-terpyridine-6,6″diyl]bis(methylenenitrilo)tetrakis(acetate)-Eu^3+^ (ATTA-Eu^3+^). In contrast to common fluorescein-based probes, phosphorescence probes and especially ATTA-Eu^3+^ could be quite useful in detecting ^1^O_2_ in weakly acidic, neutral, and basic buffers, since its phosphorescence signal is stable at pH > 3 [58]. Furthermore, its low limit of detection (LOD) of about 2.8 nmol/L, and the intense red shift of the phosphorescence signal allows the investigation of very low yields of ^1^O_2_ produced by photo-sensitizers [59]. In addition to common spectroscopic techniques mentioned above, one can also use more complex detection methods, depending on the available equipment like electron paramagnetic resonance spectroscopy (EPR) [60], nuclear magnetic resonance spectroscopy (NMR) [61] or liquid chromatography [62]. Especially EPR is a useful technique in detecting and identifying ROS, however, it is not a standard technique in the pharmaceutical industry. EPR is a spectroscopic method, which based on the detection of unpaired electrons and therefore, compounds that have a permanent magnetic moment. Most ROS possess unpaired electrons or induce radical formation that can be detected via EPR. Furthermore, very reactive, and thus, short-lived ROS can be detected by EPR via using spin-trapping techniques. Spin-trapping involves the addition of radicals to nitrone spin traps to form a spin adduct which has a relatively longer half-time to allow the detection. Analogues to EPR, NMR and chromatographic techniques also require the usage of suitable probes or chemical traps. In essence, chemical traps are used, because they change their physicochemical properties due to the reaction with ^1^O_2_, like the magnetic moment of 2,2,6,6-Tetramethylpiperidine (TEMP) [60] and thus allow the detection of low ^1^O_2_ yields. Furthermore, the usage of chemical traps allows to expand the analysis time of ^1^O_2_ from its original half-life of a few microseconds to a few minutes. In general, a broad variety of probes and chemical traps for ^1^O_2_ detection are currently available that differs in their specificity, reactivity and thermal stability [56,62,63,64,65,66]. Consequently, one must always evaluate and choose suitable spectroscopic probes or chemical traps depending on the respective question at hand.

### 3.2. Superoxide Anions (·O_2_^−^)

*Superoxide anions* (·O_2_^−^) (Table 2) are generated by single electron transfer reactions within the type I photo-oxidation pathway by photo-sensitizers such as riboflavin [67,68]. This particular ROS species can act as reductant or oxidant with redox potentials of −0.16 V regarding O_2_/·O_2_^−^ [69] or 0.94 V in the case of ·O_2_^−^, 2H^+^/H_2_O_2_ [37], but due to its overall negative charge it is a reducing agent. Hence, ·O_2_^−^ shows only a limited reactivity towards electron-rich molecules [70], which might be explaining its relative long lifetime of ≈50 µs [41]. For instance, typical oxidation reaction rates for ·O_2_^−^ towards free amino acids are approximately 1 M^−1^ s^−1^ [71]. In essence, the transfer of an additional electron to ·O_2_^−^ is not favorable, making this ROS species a better reducing than an oxidizing agent as stated previously [72]. Hence, typical targets of ·O_2_^−^ oxidation are iron-sulfur clusters or other metal ions with loose electrons, resulting in the formation of hydrogen peroxide (H_2_O_2_) [70]. Furthermore, as a reducing agent ·O_2_^−^ is also capable to reduce Fe^3+^ back to Fe^2+^, thereby facilitating the conversion of H_2_O_2_ to hydroxyl radicals (·OH) via a Fenton/Haber-Weiss reaction cycle [73,74,75]. Consequently, since ·O_2_^−^ molecules are usually not directly involved in oxidation processes but are often the precursors of other ROS with significantly higher redox potentials (Figure 2), their investigation is beneficial in understanding, e.g., potential root causes of photo-oxidation processes. The ·O_2_^−^ molecule can be monitored directly via UV-Vis spectroscopy, since it absorbs light below 300 nm, with a maximum at 245 nm in aqueous solutions [76]. Although the molar extinction coefficient ε_245_ is approximately 2350 M^−1^∙cm^−1^ [76], it is still quite challenging to detect small yields of ·O_2_^−^, especially if other compounds are present in the sample, like proteins or buffer components, which adsorption outweigh or overlay with the ·O_2_^−^ absorption. Consequently, almost all studies regarding the measurement of ·O_2_^−^ molecules utilize reactive probes or chemical traps, as previously described for detecting ^1^O_2_ (for more information, see Section 3.1). A very common approach to identify the formation of ·O_2_^−^ is chemiluminescence, since this technique is easy to operate, and a broad variety of ·O_2_^−^ reactive probes are available. Although chemiluminescence is usually measured without an additional excitation light source, it is still possible to investigate photo-chemical processes with this kind of technique, such as the photo-sensitized chemiluminescence [77] of lucigenin by the photo-sensitizer rose bengal [78]. Hence, the investigation of the formation of ·O_2_^−^ using chemiluminescence is possible, certain aspects need to be considered. Although chemiluminescence based methods are usually considered to be sensitive, the low quantum yield (φ_F_ = 0.001–0.1) of common ·O_2_^−^ probes, like luminol or lucigenin [79] lead to overall high LOD values. Thus, only reactions producing high yields of ·O_2_^−^ can be detected. Furthermore, light emitted by the light source that is required to initiate photo-chemical reactions might result in the measurement of artefacts, e.g., via scattering [80]. This is especially problematic, if the emission spectra of the chemiluminescent probe overlaps with the excitation spectra of the photo-sensitizer or the light source’s emission spectra. Furthermore, common probes are not specific for ·O_2_^−^ [81], thus without performing additional experiments, it is not possible to distinguish whether ·O_2_^−^ or other radicals are responsible for the observed chemiluminescence. In essence, chemiluminescence methods are powerful and established tools in studying oxidation processes involving ·O_2_^−^ in the absence of light, but in the context of photo-oxidation, other methods such as fluorescence-based techniques are more suitable to investigate the formation of ·O_2_^−^. In contrast to chemiluminescence, fluorescence-based techniques do not require an immediate observation of the reaction between ·O_2_^−^ and a probe, since only the resulting modification of the probe is the basis for ·O_2_^−^ detection, as previously described for ^1^O_2_. Hence, the modification of the probe and its detection can be achieved in two consecutive steps. Several fluorescence probes are known that interact with ·O_2_^−^, like dihydroethidium (DHE) or its derivative mitoSOX©. Both probes react with ·O_2_^−^ to generate the specific red fluorescent 2-hydroxyethidium (2-OH-E^+^), but are also readily oxidized by other ROS, such as ·OH or H_2_O_2_, leading to non-specific red fluorescent ethidium (E^+^) [82]. Furthermore, since the fluorescence spectra of 2-OH-E^+^ and E^+^ are very similar, it is not possible to identify which ROS is responsible for the increasing in fluorescence. Although this particular problem can be solved by utilizing additional analytical techniques, including HPLC [83] or LC-MS [84] to separate the peaks prior to analyze the spectra of 2-OH-E^+^ and E^+^, the observation that 2-OH-E^+^ decomposes to E^+^, due to visible light exposure [83], makes its putative specificity towards ·O_2_^−^, non-existent in photo-oxidation studies. Therefore, other fluorescence probes are better suited like the commercially available TEMPO-9-Ac [85,86], which contains a stable nitroxide radical (TEMPO), conjugated to a fluorescent acridine moiety. The fluorescence of the molecule is quenched by its nitroxide radical, until it is oxidized by ROS or other radicals [87,88]. In addition to TEMPO-9-Ac, other fluorescence probes are also known from literature, like HO-1889NH [89], 2-(2-thienyl)benzothiazoline (TBT) [90], or DBZTC [91], showing higher specificities, especially towards ·O_2_^−^. Alternatively, simple colorimetric assays, based on cytochrome c (cytc) or nitro blue tetrazolium (NBT), are commonly used in detecting ·O_2_^−^ formation in vivo and in vitro. In a cytc assay the reduction of ferri- to ferro-cytochrome c is observed [92], and although cytc can be reduced or even re-oxidized by various ROS [93], one can include superoxide dismutase (SOD) to demonstrate the presence of ·O_2_^−^ [94]. The same is true for the NBT assay [95,96] in which NBT is reduced to its deep-blue diformazan form by ·O_2_^−^ or other electron donors [97]. A major drawback of NBT is that its diformazan is insoluble in water, and its precipitation makes UV-Vis measurements inconsistent. Thus, the assay was optimized several times [98,99,100], for example by developing novel NBT derivatives with higher specificity towards ·O_2_^−^ and water soluble diformazan reaction products [101]. Despite their complexity in sample preparation, data interpretation and high costs, EPR [102] and NMR [103] are two expressive techniques often used in studying ·O_2_^−^. Although ·O_2_^−^ could be directly detected by both methods, the short lifetime of ·O_2_^−^ usually requires the use of spin traps [104], like DMPO [105,106] or BMPO [107]. DMPO and BMPO generate stable adducts with half-lives of approximately 1 min for DMPO-OOH and 22 min for BMPO-OOH [108], thus are the most commonly used spin traps for investigating ·O_2_^−^. Furthermore, due to the characteristic EPR and NMR spectra, both adducts can be distinguished from adducts, generated by other ROS, like hydroxyl radicals (·OH) [108]. Depending on the aim of the study, one should consider including superoxide dismutase SOD in the experiments to exclude or confirm the actual formation of ·OH, since DMPO-OOH and BMPO-OOH can decompose to their corresponding ·OH adducts [109]. Additionally, other chemical or spin traps without such specificity are also known, such as the ·O_2_^−^ scavenger Tiron [110] which is oxidized non-specifically by electron donors [111] to its semiquinone radical. These and other quinone based chemical traps are useful in detecting photo-induced radical species [112,113] not primarily via EPR or NMR, but rather via chromatography [112], whereby the extent of modification induced via ·O_2_^−^ can be evaluated by SOD [94,111].

*Superoxide anions* (·O_2_^−^) (Table 2) are generated by single electron transfer reactions within the type I photo-oxidation pathway by photo-sensitizers such as riboflavin [67,68]. This particular ROS species can act as reductant or oxidant with redox potentials of −0.16 V regarding O_2_/·O_2_^−^ [69] or 0.94 V in the case of ·O_2_^−^, 2H^+^/H_2_O_2_ [37], but due to its overall negative charge it is a reducing agent. Hence, ·O_2_^−^ shows only a limited reactivity towards electron-rich molecules [70], which might be explaining its relative long lifetime of ≈50 µs [41]. For instance, typical oxidation reaction rates for ·O_2_^−^ towards free amino acids are approximately 1 M^−1^ s^−1^ [71]. In essence, the transfer of an additional electron to ·O_2_^−^ is not favorable, making this ROS species a better reducing than an oxidizing agent as stated previously [72]. Hence, typical targets of ·O_2_^−^ oxidation are iron-sulfur clusters or other metal ions with loose electrons, resulting in the formation of hydrogen peroxide (H_2_O_2_) [70]. Furthermore, as a reducing agent ·O_2_^−^ is also capable to reduce Fe^3+^ back to Fe^2+^, thereby facilitating the conversion of H_2_O_2_ to hydroxyl radicals (·OH) via a Fenton/Haber-Weiss reaction cycle [73,74,75]. Consequently, since ·O_2_^−^ molecules are usually not directly involved in oxidation processes but are often the precursors of other ROS with significantly higher redox potentials (Figure 2), their investigation is beneficial in understanding, e.g., potential root causes of photo-oxidation processes. The ·O_2_^−^ molecule can be monitored directly via UV-Vis spectroscopy, since it absorbs light below 300 nm, with a maximum at 245 nm in aqueous solutions [76]. Although the molar extinction coefficient ε_245_ is approximately 2350 M^−1^∙cm^−1^ [76], it is still quite challenging to detect small yields of ·O_2_^−^, especially if other compounds are present in the sample, like proteins or buffer components, which adsorption outweigh or overlay with the ·O_2_^−^ absorption. Consequently, almost all studies regarding the measurement of ·O_2_^−^ molecules utilize reactive probes or chemical traps, as previously described for detecting ^1^O_2_ (for more information, see Section 3.1). A very common approach to identify the formation of ·O_2_^−^ is chemiluminescence, since this technique is easy to operate, and a broad variety of ·O_2_^−^ reactive probes are available. Although chemiluminescence is usually measured without an additional excitation light source, it is still possible to investigate photo-chemical processes with this kind of technique, such as the photo-sensitized chemiluminescence [77] of lucigenin by the photo-sensitizer rose bengal [78]. Hence, the investigation of the formation of ·O_2_^−^ using chemiluminescence is possible, certain aspects need to be considered. Although chemiluminescence based methods are usually considered to be sensitive, the low quantum yield (φ_F_ = 0.001–0.1) of common ·O_2_^−^ probes, like luminol or lucigenin [79] lead to overall high LOD values. Thus, only reactions producing high yields of ·O_2_^−^ can be detected. Furthermore, light emitted by the light source that is required to initiate photo-chemical reactions might result in the measurement of artefacts, e.g., via scattering [80]. This is especially problematic, if the emission spectra of the chemiluminescent probe overlaps with the excitation spectra of the photo-sensitizer or the light source’s emission spectra. Furthermore, common probes are not specific for ·O_2_^−^ [81], thus without performing additional experiments, it is not possible to distinguish whether ·O_2_^−^ or other radicals are responsible for the observed chemiluminescence. In essence, chemiluminescence methods are powerful and established tools in studying oxidation processes involving ·O_2_^−^ in the absence of light, but in the context of photo-oxidation, other methods such as fluorescence-based techniques are more suitable to investigate the formation of ·O_2_^−^. In contrast to chemiluminescence, fluorescence-based techniques do not require an immediate observation of the reaction between ·O_2_^−^ and a probe, since only the resulting modification of the probe is the basis for ·O_2_^−^ detection, as previously described for ^1^O_2_. Hence, the modification of the probe and its detection can be achieved in two consecutive steps. Several fluorescence probes are known that interact with ·O_2_^−^, such as dihydroethidium (DHE) or its derivative mitoSOX©. Both probes react with ·O_2_^−^ to generate the specific red fluorescent 2-hydroxyethidium (2-OH-E^+^), but are also readily oxidized by other ROS, such as ·OH or H_2_O_2_, leading to non-specific red fluorescent ethidium (E^+^) [82]. Furthermore, since the fluorescence spectra of 2-OH-E^+^ and E^+^ are very similar, it is not possible to identify which ROS is responsible for the increasing in fluorescence. Although this particular problem can be solved by utilizing additional analytical techniques, including HPLC [83] or LC-MS [84] to separate the peaks prior to analyze the spectra of 2-OH-E^+^ and E^+^, the observation that 2-OH-E^+^ decomposes to E^+^, due to visible light exposure [83], makes its putative specificity towards ·O_2_^−^, non-existent in photo-oxidation studies. Therefore, other fluorescence probes are better suited such as the commercially available TEMPO-9-Ac [85,86], which contains a stable nitroxide radical (TEMPO), conjugated to a fluorescent acridine moiety. The fluorescence of the molecule is quenched by its nitroxide radical, until it is oxidized by ROS or other radicals [87,88]. In addition to TEMPO-9-Ac, other fluorescence probes are also known from literature, such as HO-1889NH [89], 2-(2-thienyl)benzothiazoline (TBT) [90], or DBZTC [91], showing higher specificities, especially towards ·O_2_^−^. Alternatively, simple colorimetric assays, based on cytochrome c (cytc) or nitro blue tetrazolium (NBT), are commonly used in detecting ·O_2_^−^ formation in vivo and in vitro. In a cytc assay the reduction of ferri- to ferro-cytochrome c is observed [92], and although cytc can be reduced or even re-oxidized by various ROS [93], one can include superoxide dismutase (SOD) to demonstrate the presence of ·O_2_^−^ [94]. The same is true for the NBT assay [95,96] in which NBT is reduced to its deep-blue diformazan form by ·O_2_^−^ or other electron donors [97]. A major drawback of NBT is that its diformazan is insoluble in water, and its precipitation makes UV-Vis measurements inconsistent. Thus, the assay was optimized several times [98,99,100], for example by developing novel NBT derivatives with higher specificity towards ·O_2_^−^ and water soluble diformazan reaction products [101]. Despite their complexity in sample preparation, data interpretation and high costs, EPR [102] and NMR [103] are two expressive techniques often used in studying ·O_2_^−^. Although ·O_2_^−^ could be directly detected by both methods, the short lifetime of ·O_2_^−^ usually requires the use of spin traps [104], such as DMPO [105,106] or BMPO [107]. DMPO and BMPO generate stable adducts with half-lives of approximately 1 min for DMPO-OOH and 22 min for BMPO-OOH [108], thus are the most commonly used spin traps for investigating ·O_2_^−^. Furthermore, due to the characteristic EPR and NMR spectra, both adducts can be distinguished from adducts, generated by other ROS, such as hydroxyl radicals (·OH) [108]. Depending on the aim of the study, one should consider including superoxide dismutase (SOD) in the experiments to exclude or confirm the actual formation of ·OH, since DMPO-OOH and BMPO-OOH can decompose to their corresponding ·OH adducts [109]. Additionally, other chemical or spin traps without such specificity are also known, such as the ·O_2_^−^ scavenger Tiron [110] which is oxidized non-specifically by electron donors [111] to its semiquinone radical. These and other quinone based chemical traps are useful in detecting photo-induced radical species [112,113] not primarily via EPR or NMR, but rather via chromatography [112], whereby the extent of modification induced via ·O_2_^−^ can be evaluated by SOD [94,111].

### 3.3. Hydrogen Peroxide (H_2_O_2_)

*Hydrogen peroxide* (H_2_O_2_) (Table 3) is a secondary product, generated via the type I photo-oxidation pathway by electron transfer [114,115] to ·O_2_^−^ or spontaneous disproportionation of ·O_2_^−^ via the hydroperoxyl radical (HOO·) degradation pathway [116]. Since the H_2_O_2_ molecule has a reaction rate below < 10^2^ M^−1^∙s^−1^ [18], its reduction back to ·O_2_^−^ is not favored (E^0^(·O_2_^−^, 2H^+^/H_2_O_2_) = 0.94 V) and its oxidation potential towards ·OH is rather low (E^0^(H_2_O_2_, H^+^/H_2_O,·OH) = 0.32 V) [37]. The combination of its low redox potentials and the presence of a stable O-O bond might explain why H_2_O_2_ molecules exhibit a fairly long lifetime of hours at moderate conditions (acidic pH and physiological temperature) [117]. Nevertheless, in the presence of metal ions such as Fe(II), H_2_O_2_ can be converted via the Haber-Weiss reaction to the strongly oxidizing ·OH radical [75]. Therefore, despite its low reactivity, H_2_O_2_ needs to be considered as an important intermediate in ROS mediated oxidation pathways. Consequently, several techniques are available for H_2_O_2_ detection. Similar to other ROS, H_2_O_2_ can be directly measured via UV-Vis spectroscopy. However, due to its low extinction coefficient of 43.6 M^−1^ cm^−1^ at 240 nm [118] this approach is just suitable to determine high H_2_O_2_ concentrations (35% (*w*/*w*)) [119] of stock or calibration solutions. Hence, investigating the formation of low H_2_O_2_ yields during photo-oxidation processes requires methods with higher sensitivity. On an analytical aspect, the advantage of H_2_O_2_ is its long lifetime [117], which offers the possibility to “delay” H_2_O_2_ detection after sample irradiation. Although this makes the investigation of H_2_O_2_ formation experimentally less sophisticated, one must consider that light is capable to degrade H_2_O_2_ to ·OH via photolysis [120,121,122]. Therefore, H_2_O_2_ yields determined after light irradiation might not reflect the actual H_2_O_2_ formation during light exposure. Photolysis of H_2_O_2_ follows a first order kinetic and is most prominent for UV light [123] and only minor for visible light [120,121]. The magnitude of photolysis can be determined by including control samples with known H_2_O_2_ concentrations in the experimental set up. Thus, subsequent analysis can still be considered as a valid approach in investigating H_2_O_2_ formation. Consequently, techniques commonly used for H_2_O_2_ analysis can be utilized for analyzing H_2_O_2_ formation in the context of photo-oxidation, however, none of the analytical techniques discussed in the subsequent section are capable to measure H_2_O_2_ with high specificity. This is not only because of the mentioned disproportionation process. In this regard, changes in signal intensity, induced by H_2_O_2_, can be distinguished from other oxidizing agents by utilizing the enzyme catalase. Since catalase converts H_2_O_2_ to water and oxygen [124], differences in signal intensity between treated and untreated samples yield the actual amount of generated H_2_O_2_ [125]. For monitoring H_2_O_2_ yields, the most common approach is the use of horseradish peroxidase-based fluorometric assays. In this type of assay the horseradish peroxidase (HRP) use H_2_O_2_ to oxidize fluorogenic substrates such as Amplex^®^ Red (N-acetyl-3,7-dihydroxyphenoxazine) stoichiometrically [126] to highly fluorescent products such as resorufin [127]. The high sensitivity of fluorescence based techniques allow the measurement of H_2_O_2_ concentrations as low as 50 nM in the case of Amplex^®^ Red [126] without interferences of most other sample components. In this regard, the use of simpler HRP-based colorimetric assays with substrates such as TMB (3,5,3′,5′-tetramethylbenzidine) can be considered, if no or only minor interference of absorbing sample components are expected [128,129]. It must be noted that monitoring H_2_O_2_ formation during sample irradiation by HRP-based assays is not recommended, since most dyes or probes are either photo-sensitive or ROS producing photo-sensitizers, which might result in invalid measurements and/or artefacts [130]. In contrast, other colorimetric methods require rather harsh conditions and thus are primarily suited for investigating H_2_O_2_ yields after irradiation. However, these assays are easy to handle, sensitive, and robust, therefore are broadly used in measuring H_2_O_2_ such as the FOX assay. The FOX assay is based on the oxidation of ferrous ions (Fe^2+^) to ferric ions (Fe^3+^) in acidic conditions by H_2_O_2_. The subsequent reaction of the formed Fe^3+^ with xylenol orange creates a blue-purple complex allowing the detection of peroxides as low as 100 nM [131]. Furthermore, the FOX assay is quite specific for peroxides including hydrogen or organic peroxides [125,131,132]. In addition to pure spectroscopic methods, chemical traps are also used for H_2_O_2_ analysis via chromatography such as triphenylphosphine (TPP) [133]. Similar to the FOX assay, TPP is predominantly oxidized by peroxides to triphosphine oxide (TPPO) [125,134] with an average detection limit of approximately 1 µM for H_2_O_2_ [133]. Unfortunately, TPP cannot be used to detect H_2_O_2_ formation during light irradiation, since TPP is light sensitive.

### 3.4. Hydroxyl Radicals (·OH)

*Hydroxyl radicals* (·OH) (Table 4) can be generated in the later stages of photo-oxidation from H_2_O_2_ via the Haber-Weiss reaction mechanism [75] as previously mentioned. The ·OH radical is the most reactive ROS with an average half-life of 1 ns and rate constants near the diffusion limit of 10^9^–10^10^ M^−1^∙s^−1^ [135]. Furthermore, ·OH radicals are strong oxidizing agents with a redox potential of 2.3 V (·OH, H^+^/H_2_O) [37]. Therefore, ·OH radicals are rather unspecific, reacting with a broad variety of molecules via electron transfer, hydrogen abstraction and double bond addition [135,136], resulting in the formation of other highly reactive molecules such as organic radicals (R·) [137] and their corresponding peroxide radicals (ROO·) [138,139]. The ·OH radicals react with solvent accessible amino acid side chains, especially hydrophobic or sulfur-containing residues, resulting in a variety of side chain modifications [140], in particular a mass addition of +16 Da (+O), +32 Da (+2O) and +14 Da (CH2 to CO) [141,142,143] can often be detected using LC/MS. Hence, various analytical methods were developed to further characterize the reactivity and hazardousness of ·OH radicals. The short lifetime and low extinction coefficient of 665 M^−1^∙cm^−1^ at 230 nm [144] pose a challenge regarding the direct detection of ·OH in aqueous environments. Consequently, formation of ·OH is usually monitored indirectly via observing ·OH-induced modifications of spectroscopic probes or chemical traps [145,146]. Interestingly, most probes and chemical traps for ·OH detection exploit the characteristic ability of ·OH radicals to hydroxylate aromatic residues [147,148,149] which explains their high specificity for this kind of ROS. In this regard, a well-established fluorogenic probe is the water-soluble terephthalic acid (TPA) which is converted to the fluorescent 2-hydroxyl terephthalic acid (HTPA) by ·OH radicals [150,151]. The high scavenging rate constant of 3.3 × 10^9^ M^−1^∙s^−1^ for ·OH of TPA [152], together with the spectral properties of HTPA (ε_210_ = 28,400 M^−1^∙cm^−1^; φ_F_ = 0.65) [153], allows the detection of ·OH down to 5 nM [154]. Furthermore, TPA does not react to a greater extent with other ROS besides ·OH and is insensitive to light [153,155], however, HTPA shows signs of photo-degradation if exposed to light below 400 nm [153]. Consequently, depending on the light source and exposure time, TPA can be utilized to study ·OH formation during photo-oxidation processes. Nevertheless, other fluorogenic probes, like 3′-(p-Aminophenyl) fluorescein (APF) or 3′-(p-Hydroxyphenyl) fluorescein (HPF), that possess similar spectral properties, specificities, and detection limits regarding ·OH, but with better photo-stabilities are also available [156,157,158]. Additionally, TPA, APF or HPF and their respective oxidation products can be further quantified by using chromatographic techniques, if certain components present in the sample might interfere with their fluorescent signal [154]. On this note, by using HPLC a broad variety of chemical traps for ·OH detection is available, such as salicylic acid or phenylalanine [159,160,161]. Although these HPLC-based techniques show equal limits of detection for ·OH in the lower nanomolar range like 20 or 50 nM for salicylic acid [159] or phenylalanine [161] respectively, they possess certain drawbacks. For instance, more expensive and sophisticated equipment is required, compared to standard fluorescence readers. Furthermore, the chromatograms are often difficult to interpret, since the chemical modification of these traps results in the formation of multiple ortho-, meta-, and para-oxidation products [161]. Hence, depending on the experimental conditions and the respective research topic under investigation, between both techniques fluorogenic traps should be the first choice for investigating ·OH formation. Alternatively, EPR in combination with spin traps like DMPO [162] or BMPO [163] is a very common approach in detecting ·OH formation as mentioned in the context of ·O_2_^−^. Although, the hydroxylation adducts of both spin traps can be clearly identified by their EPR spectra [108], their half-lives of approximately 14 to 25 or 30 min for DMPO-OH [164,165] or BMPO-OH [163] makes quantification of ·OH generation quite challenging for long-lasting experiments. Furthermore, peroxyl adducts created via ·O_2_^−^ can spontaneously decompose to their corresponding hydroxylation products [109] resulting in deviations for ·OH quantification, similar to those previously discussed for ·O_2_^−^.

### 3.5. Carbon-Centered Radicals (R·), Alkoxyl Radicals (RO·), Peroxyl Radicals (ROO·), and Organic Hydroperoxides (ROOH) 

Carbon-centered radicals (R·), alkoxyl radicals (RO·), peroxyl radicals (ROO·), and organic hydroperoxides (ROOH) are heterogenous molecule classes, occasionally produced by ROS such as ·OH. Although, these molecules can be clearly assigned to their respective class, however within each class, molecules can still significantly differ in relevant physicochemical properties due to their different origins [166,167,168]. Hence, it is not feasible, within the frame of this review, to summarize typical methods for detecting all members of each class. For detailed species information see Davies et al. (2004) [169]. Nevertheless, specific techniques for investigating important degradation products of therapeutic protein formulations, like the autooxidation of polysorbates, will be discussed in detail in Section 5. Therefore, hereinafter only some general characteristics of these molecule classes are briefly reviewed. Carbon-centered radicals (R·) are generated by proton abstraction via other radical species such as ·OH (Figure 3), thereby initializing the subsequent formation of the previously mentioned organic oxygen-centered radicals or hydroperoxides [137,170,171]. The R· and RO· radicals are highly reactive towards other organic molecules with rate constants of up to 10^9^ M^−1^∙s^−1^ [172] and 10^7^ M^−1^∙s^−1^ [173], whereas ROO· and ROOH show rate constants below 10^2^ M^−1^∙s^−1^ [173]. A similar tendency can be observed regarding their redox potentials, with R· and RO· being strong oxidizing agents possessing E^0^ values of 1.9 and 1.6 V, while the oxidizing capabilities of ROO· and ROOH are moderate in the range of 0.77 to 1.44 V [37]. Consequently, ROOH molecules are rather stable with half-lives ranging from minutes up to several hours [174,175,176]. In contrast, typical half-lives for R· and RO· radicals are 10^−8^ and 10^−6^ s and ROO· being an intermediate species with 7 s [177].

## 4. Photo-Oxidation of Proteins via ROS and Detection of Specific Modifications

The formation of different ROS by UV and ambient light via energy or electron transfer reactions [178] as described, lead to the oxidation of various organic molecules, such as proteins in many different chemical and biological systems. As stated frequently in literature, amino acids can absorb UV light which might damage the proteins. In comparison to UV light, in the visible range none of the proteinogenic amino acids absorb light, which leads to an unclear mechanism [179]. In accordance with Schöneich (2020), compounds such as impurities from manufacturing process or additives may act as photosensitizers. These non-proteinogenic photosensitizers absorb visible light and produce ROS that can in turn damage other molecules, including proteins. In essence, although the UV and visible area photooxidation mechanisms start with the excitation of a photosensitizer, the degradation process in the visible area requires ROS species as mediators for the subsequent chain reactions. The mechanisms of the oxidation reactions differ for each ROS, however, highly reactive radical species, such as ·OH or RO· create a broad variety of derivates and thus, possess the greatest hazardous potential. These radicals typically undergo a variety of reactions, such as addition, fragmentation, rearrangement, dimerization, disproportionation and substitution, but most reactions with proteins take place through three different reaction mechanisms: (a) hydrogen atom abstraction from C–H, S–H, N–H and O–H; (b) electron abstraction from electron rich sites, such as those of His, Trp and (c) addition to electron-rich centers, such as aromatic rings or sulfur species [178]. In essence, ROS react either with the protein backbone [180,181] or primarily the amino acid side chains [178] leading to global changes in the protein structure, such as fragmentation [177], partial unfolding [182], aggregation [183,184] or cross-linkages [185,186,187]. ROS, such as ·OH radicals are very reactive and react rapidly with surface accessible amino acid side chains (further information see Section 3.4) [140]. Very strong oxidizing agents can result in localized free radical formation which further can oxidize the protein backbone by producing α-carbon radicals (further information is given in Section 4.1). Furthermore, each oxidative amino acid residue modification can change the physical and chemical properties of the entire protein up to its functional loss and biological inactivation [2,12,186,188].

### 4.1. Structural Perturbations Due to Photo-Oxidation

*Fragmentation processes* on the backbone of mAbs induced by light are less often documented for protein drugs than side chain modifications [169,178,181]. However, some ROS are known to fragment the backbone [189] as a result of their conformational flexibility and solvent conditions [180]. Especially fragmentation around the hinge region has been documented [190,191]. A well-known oxidative “attack” on the protein backbone is the abstraction of hydrogen atom from the α-carbon, resulting in the formation of α-carbon-centered radical [178,192] as previously described and depicted in Figure 3. The following radical species can undergo electron transfer with acceptors such as O_2_, disulfides or His residues [193]. The reaction with oxygen leads to the formation of peroxyl radicals or alkyloxyradicals [194,195]. In addition, side chain oxidation can undergo radical transfer to induce backbone oxidation and vice versa [178,196]. Finally, further reaction mechanism(s) can lead to fragmentation of the protein backbone [136,192,197].

Analytical methods detecting fragmentation processes can be subdivided into two groups according to their mode of separation. The first group of separation methods deals with the hydrodynamic size of protein modifications such as size-exclusion chromatography (SEC), sodium dodecyl sulfate-polyacrylamide gel electrophoresis (SDS-PAGE) and capillary electrophoresis (CE) with SDS. Further information is given in later paragraphs. The second separation group is based on the chemistry of aggregates, fragments, and the protein, including different types of chromatography, such as hydrophobic-interaction chromatography (HIC) [198,199], cation-exchange chromatography (CEX) or reversed-phase chromatography (RP). The methods based on size are used mainly to monitor and quantify the degraded samples, while HPLC methods in combination with mass spectrometry (MS) are mainly used to identify the exact cleavage sites [180]. Methods based on size can only detect peptide bond cleavage if the molecule is separated into two fragments, since under native conditions non-covalent interactions and disulfides prevent separation of the two fragments [200]. Therefore, SEC can be used to quantify the extent of fragmentation under native conditions, especially hinge fragmentation [8,201]. SEC can be performed under denaturing conditions [200,202], so that cleavage of e.g., mAb domains can be detected in combination with reduction of the disulfide bonds. SDS-PAGE and CE-SDS can be used to monitor overall fragmentation [203,204]. Especially CE-SDS is used in pharmaceutical industry due to the straightforward quantification and better resolution compared to the traditional slab gel SDS PAGE [180], and the opportunity to improve the sensitivity with fluorescence detection to detect fragmentation patterns [203,204]. But the identification of cleavage sites is complicated due to its difficulty to collect the fractions [205,206]. The exact identification of fragmentation sites is carried out by HPLC methods coupled with MS (LC-MS) or N-terminal sequencing [207,208]. HPLC methods, such as hydrophobic-interaction chromatography (HIC) [198,199] or cation-exchange chromatography (CEX) [209] is used to detect peptide bond cleavage but are commonly applied to detect degradations of amino acid side chains [180]. Most information about fragmentation can be achieved by coupling reversed phase (RP) HPLC with in-line MS [200,210]. Fragments from amino acid side-chain degradation can co-elute in RP-HPLC-MS, so that the technique should not be used to monitor overall fragmentation [180]. However, tandem MS approaches can separate fragments with similar Dalton numbers, so that they become recognizable. Furthermore, hydrogen deuterium exchange mass spectrometry (HDX-MS), is a well-established method in the field of structural MS to provide the solvent accessibility, dynamics and hydrogen bonding of backbone amides in proteins [211].

*Protein aggregation* rates can be altered via conformational changes and fragmentation, which may influence the quality and function of mAbs [180,212]. In contrast to the limited evidence for backbone cleavage and fragmentation of photo-oxidized mAbs, the formation of high molecular weight aggregates (dimers and higher oligomeric species) has been reported in numerous studies [7,20,191,213,214,215,216]. Aggregate formation can occur for example through unfolding by fragmentation or through side chain modification. For further information see Lévy et al. (2019) [217]. Aggregates can be distinguished between noncovalent and covalent species, and the aggregation state can be reversible or irreversible. Gross-Rother et al. (2020) have recently reviewed various analytical techniques used to determine protein aggregation [218]. SEC, which based on the hydrodynamic size of proteins, is used for analysis and quantification of soluble aggregates that are noncovalent and irreversible. Reducing SEC can distinguish covalent aggregates from intact antibodies [202] by monitoring the decreased total content via reduction of disulfide bridges. Although SEC offers high-throughput efficiency, its analytical accuracy is limited by interactions of the protein with the matrix. Moreover, the detectable hydrodynamic size range is limited [219], because larger aggregates can be filtered out by frits in the system or by the column itself. Consequently, large protein aggregates may disappear and can be overlooked in analysis. Another form of aggregates that may be not detected during SEC are very low affinity intermolecular association, as these may dissociate into monomers, e.g., during dilution in the SEC column via mobile phase or change in temperature. To identify low affinity aggregates other SEC method conditions should be adapted or techniques, such as analytical ultracentrifugation (AUC) can be used. Since photo-oxidation can lead to cross-linking products [2,183], techniques have been applied to separate intermolecular cross-links, such as di-tyrosine and cystine by SEC [220,221] or SDS-PAGE. SEC is beneficial as it avoids in-gel digestion, which may be inefficient for cross-linked species [222]. With denaturing SEC, a differentiation between covalent and non-covalent residues is possible. SDS-PAGE can be used to detect covalent aggregates (HMWs, high molecular weight species) in photo-oxidized samples [223]. When performed under reducing conditions, the method separates aggregates bound by disulfides from other non-reducible covalent bonds [224,225]. SDS-PAGE is becoming replaced by CE-SDS, as the latter has a higher resolution and better robust quantification. Since CE-SDS cannot detect noncovalent aggregates, the pharmaceutical industry combines SEC with SDS-PAGE and/or CE-SDS. Covalent aggregate formation can be distinguished between intermolecular and intramolecular species. The low abundant intermolecular species can be enriched prior to analysis by strong cation exchange chromatography, as these species have a larger number of protonable sites [226]. AUC in sedimentation velocity mode is used to characterize the sedimentation behavior of soluble aggregates in solution. In contrast to SEC or SDS-PAGE, there is only little to no sample preparation necessary, depending on protein concentration. A high protein concentration hinders a reliable molecular weight determination, so that dilution must be carried out with potential artefact generation. Spectroscopic methods, such as fluorescence spectroscopy [8,227,228], circular dichroism (CD) [229], NMR [227,230,231,232], Fourier transformation infrared spectroscopy (FTIR) [232,233,234,235,236] and Raman spectroscopy [229] can be used to determine conformational changes, so that aggregation states can be recorded [237]. These methods can distinguish an intact protein from an aggregated protein, if isolated. However, since proteins have different aggregate morphology, they overlap in the signals and therefore resolution may be impaired. Separation techniques, such as chromatography and electrophoresis [238] are combined with detectors, such as UV, fluorescence and MS. LC-MS techniques have been employed to detect and quantitate sites of oxidation in protein samples [239] as well as aggregation [240,241]. Three fundamental strategies are employed to detect oxidized proteins as well as aggregates. The “top down” approach monitors intact protein molecular ions, which are generated by electrospray-ionization (ESI) or matrix-assisted laser desorption/ionization (MALDI) and then introduced into the mass analyzer [242]. This “top down” approach offers the advantage of providing access to the complete primary protein sequence and the ability to characterize the total load of post-translational modifications [243]. The technique has been applied to detect methionine oxidation in filgrastim successfully [244] and monitor aggregation [245]. The “top down” approach using ESI-MS is to produce multiple charged ions during ionization, which generate complex mass spectra [231,243]. Therefore, this technique is often limited to isolated proteins or simple protein mixtures [231] and cannot reveal the specific amino acid site, involved in the post translational modification (PTM). The “middle-down” MS workflow enables a substantial increase in sequence coverage but requires partial proteolysis and deglycosylation of the mAb [246,247]. The “bottom-up” analysis is used for sequence determination, in which proteins are enzymatically digested into a peptide mixture before being introduced to the instrument. Nevertheless, it requires long sample preparation with the risk of introducing modifications as artefacts [248]. Since the proteins are digested, information about the location of oxidation and cross-links in the sequence can be lost. The HDX-MS using “bottom up” approach is applied to observe selective conformational changes at Trp residues after selectively oxidizing them [249]. Both gas chromatography (GC) MS and LC-MS have been used to identify and quantify cross-link species. Analysis at the peptide level usually involves proteolytic digestion, followed by LC-MS^n^ analysis, as described above. This is challenging for cross-links because linear (non-cross-linked) peptides usually constitute the majority of peptides, and the cross-linked species are easily overlooked [222]. Either the low abundant cross-species has to enriched by SEC [220], SDS-PAGE, SCX [226], charged-base fractional diagonal chromatography (ChaFRADIC) [250] multi step methods [251] or special MS workflows have to be applied. To identify low abundant cross-linked species, a MS workflow which uses ^18^O-labeling with trypsin has been applied. The approach is based on the ability of serine proteases to catalyze the incorporation of two ^18^O atoms from isotopically labelled water into the C-terminus of a catalyze proteolytic peptide. In contrast, cross-linked peptides, which have two C-termini, incorporate four ^18^O atoms. This strategy has been used successfully for the identification of cross-links in proteins. With this technique, for example, His-His, Trp-Trp, Tyr-Tyr, Tyr-Trp, Tyr-Lys, His-Arg and His-Lys crosslinks, could be identified [185,187,252,253,254,255].

### 4.2. Detection of Specific Modifications

In the following sections, we will discuss photo-oxidation related protein modifications capable to change relevant physicochemical properties of a protein, thus leading to the previously mentioned harmful fragmentation and aggregation behavior.

#### 4.2.1. Tryptophan Derivatives

*Tryptophan* (Trp) (Figure 1 and Table 5) is a well-known near-UV photo-sensitizer [188,256,257], since its indole moiety can absorb light with wavelengths of 320 nm or shorter and possess a good molar extinction coefficient of 5500 M^−1^∙cm^−1^ at 280 nm [258]. The pyrrole ring in Trp is prone to oxidation by ROS such as ·OH, ^1^O_2_, O_3_ [188,259]. The oxidation of Trp can be rapid due to its low oxidation potential and results in a wide range of derivates including ring hydroxylated compounds and ring-opened products [260]. For more information about the oxidation rates of Trp see Lemus et al. (2016) [261]. The ring opened products are effective photo-sensitizing agents. Examples are, Kynurenine (Kyn) with λ_max_ ≈ 361 nm [262], N-Formyl kynurenine (NFK) with λ_max_ ≈ 318 nm in water [262,263] and 3-Hydroxykynurenine (3-OH-Kyn) with λ_max_ ≈ 365–375 nm [188,256,264] (Figure 1). These derivates all extend more or less into the visible region of the electromagnetic spectrum [12,262,265], so that they can generate radicals in the visible area, which are further capable of damaging proteins [266,267]. In addition, Haywood et al. (2013) showed that UV light (λ ≈ 254 nm) induces a conversion of protein-bounded Trp to glycin (Gly{ XE “Gly” \t “*glycin*” }) and Gly hydroperoxides in IgG1 [268]. The conversion to an organic hydroperoxide may induce further protein oxidation upon storage [269].

The loss of the parent Trp has often been used as a marker for protein oxidation [260]. Since Trp has the highest fluorescent quantum yield among all natural amino acids, with Φ_f_ ≈ 0.12 ± 0.01 [281] in water, it can be used in fluorescence spectroscopy for investigations of protein structural changes and dynamics [270]. The spectrum can be used as “prior and post” evidence for Trp residue modifications through oxidation [271,272]. The detection of oxidation changes in proteins with fluorescence spectroscopy can be challenging due to the protein tertiary structure and neighboring amino acid residues, showing high intrinsic background fluorescence [188]. While excitation of Trp-containing proteins at 290 nm produces a strong emission with a peak at 350 nm, the application of fluorescence to identify the products of Trp oxidation is challenging, as will be explained further down. Chromatographic techniques, such as RP-HPLC, are used to separate Trp oxidation products in proteins [273]. Yang et al. (2007) published a RP-HPLC method to monitor Trp oxidation in the mAb heavy chain of protein with a low detection limit of 0.305% total area [274]. The two Trp oxidation products Kyn and NFK can be detected with fluorescence spectroscopy as well, but the lower fluorescence yields of especially Kyn and NFK in contrast to its parent Trp makes identification more difficult [188,275]. McAvan et al. (2020) monitored Trp oxidation products, such as NFK, Kyn and 3-OH-Trp with Raman and CD spectroscopy, both in combination with principal component analysis. Since structural differences between forced degradation samples are small and difficult to distinguish, the combination offers the opportunity to draw out small structural differences [229]. In Raman spectroscopy, peaks shifts have been shown in the indole of Trp, specifically at N–H vibration (ca. 885 cm^−1^), C–N vibration (ca. 1121 cm^−1^) and C-H vibration (1450 cm^−1^), indicating environmental changes such as Trp oxidation modifications [229]. In samples with peak shifts, the aggregate content was increased [229]. In CD spectroscopy combined with principal component analysis, the differences at 203 nm were indicative for a change in the antiparallel ß-sheet structure. An increase at 290 nm was suggested to be a Trp modification [229].

ESI-LCMS is a powerful method to identify oxidized Trp oxidation residues, since residues, such as Kyn, NFK and 3-OH-Kyn show a change in mass of Kyn −13 Da, NFK +15 Da, 3-OH-Kyn +4 Da in comparison to Trp [276,277]. Characterization of the sites of oxidation can be obtained by a MS “bottom up” workflow, in which Pavon et al. (2019) found an increased susceptible Trp oxidation at the complementary determining region (CDR) in three different mAbs [282]. The presence of multiple Trp oxidation products in a low quantity complicate MS spectrum analysis [12], but the problem can be solved by the selective accumulation of specific modifications [283,284,285]. A complementary approach to MS is NMR spectroscopy, which can be applied for Trp oxidation products, such as NFK and Kyn as well [230]. NMR can be used to analyze full length proteins in solution, as long as they can be denaturized [230]. Since the published NMR protocol relies on the random-coil chemical shift fingerprints of modified residues, it provides the chemical identity, but not the position of the modification [230]. An immunological approach using antiserum for the detection of NFK and KYN was developed by Staniszewska and Nagaraj (2007) [278]. The authors showed that the monoclonal antibody cross-reacted with free Kyn and NFK, but the reaction was weaker than with Kyn-modified amino acids [278]. Ehrenshaft et al. (2009) proposed to a polyclonal antiserum for the detection of NFK in proteins [279]. The advantage of using anti-NFK antiserum is a higher specificity towards NFK and a very small affinity towards Kyn or Trp [279]. Ehrenshaft (2009) showed a high sensitivity of ≤ 57 pmoles of NFK within 2.5 μg of total protein, but also a small cross-reactivity towards non-oxidized samples [279]. The polyclonal antiserum maybe used as high throughput technology. However, it has difficulties in locating NFK in proteins [286,287], as it requires gene-specific antibodies to recognize parts of the target proteins in addition to NKF.

The Trp residue can also form ditryptophan (di-Trp) products (C–C) and (C–N) under light exposure [288]. Limited reports on this type of cross-link have been found, potentially due to difficulties in identifying and quantifying all cross-linking species, as described in the previous section regarding protein aggregation. Direct spectrophotometric approaches can be used to determine di-Trp, such as the UV absorbance and fluorescence excitation (λ_ex_ ≈ 280 nm and λ_em_ ≈ 410 nm) [280]. Since other present chromophores or fluorophores affect these approaches, the protein has to be digested resulting in peptides. Hence, prior to their detection the peptides can then be separated using HPLC or GC approaches [280]. Alkaline hydrolysis is typically used for di-Trp since the indole-ring of Trp is susceptible to acidic cleavage [289]. After digestion, detection approaches such as MS can be applied to quantify di-Trp species. With the ^18^O-labelling MS workflow, the identification of di-Trp, as well as Tyr-Trp in proteins is possible [252,254].

#### 4.2.2. Tyrosine

*Tyrosine* (Tyr) (Figure 2 and Table 6) contains an aromatic moiety. This chromophore can absorb light in the UV range with λ_max_ ≈ 275 nm [3,256]. The oxidation mechanism of Tyr leads to a tyrosine phenoxyl radical, which can undergo free arrangements to form intra- and/or inter- protein-protein cross-linkages, such as dityrosine (di-Tyr) [3,290]. In a further reaction mechanism, di-Tyr can absorb light in the UV-A range [179,291,292], which leads to further production of ROS such as ·O_2_^−^, ^1^O_2_ and H_2_O_2_. Di-Tyr is often used as a common name for the carbon-carbon dimer and the carbon-oxygen bonded dimer. The carbon-carbon dimer appears to be the more important species [260]. The ionized di-Tyr chromophore, in which one of the two phenolic hydroxyl groups is dissociated, is responsible for the absorption of di-Tyr at 400 nm [293]. Direct UV absorbance measurements are used for di-Tyr detection, with λ_ex_ ≈ 280 nm [280]. Di-Tyr has an intrinsic fluorophore, which enables the measurement of the signal at 420 nm upon excitation [293]. However, spectroscopic methods are limited due to disturbing fluorophores from Trp and its degradation products. A variety of HPLC-based methods are currently used to identify and quantify di-Tyr. RP-HPLC with fluorescence [293,294,295] or electrochemical detection after proteolysis or acid digestion [293,296,297,298] can be used. Precolumn derivatization of protein with dabsyl chloride, followed by RP-HPLC, gives a complete amino acid analysis including di-Tyr [293]. Derivatization with RP-HPLC is useful for small quantities of dityrosine, when fluorescence detection systems are not available [293]. Malencik et al. (1996, 2003) have developed different procedures to isolate and analyze di-Tyr [296,299], e.g., by using affinity chromatography [293]. Affinity chromatography on immobilized phenylboronate can be used to separate di-Tyr from Tyr and other amino acids [293]. The technique is based on the association of di-Tyr with boric acid/monoborate and two-dimensional chromatography on BioGel P-2 [293]. The pH dependent chromatography on BioGel P-2 has the advantages of simplicity and applicability to both large preparative scale and small quantities of di-Tyr. The major limitation in di-Tyr research is the inability to determine the location and to identify low abundant di-Tyr in proteins. Thus, prior isolation and separation is often required to enrichment of the low levels of di-Tyr is often required. For oligomers containing intermolecular cross-links SEC [220], SDS-PAGE, SCX [226], charged-base fractional diagonal chromatography [250] and multistep methods [251], can be used to isolate cross-linked from non-cross-linked species. After enrichment free di-Tyr and di-Tyr residues in proteins can be identified with ESI-MS/MS. MS/MS spectra of cross-linked peptides is highly complex, due to the presence of fragment ions from both peptide chains [220,300,301]. Other cross-linking species such as Tyr-Trp, Tyr-Lys, Tyr-His can be monitored with MS as well. The low abundance of such species can be overcome by the use of ^18^O labeling with trypsin in MS. For further information see Mariotti et al. (2018), Degendofer et al. (2018) and Leinisch et al. (2017, 2018) [185,253,254,255]. Immunodetection methods, such as the use of antibodies, can be used to detect di-Tyr [302,303]. They are semi quantitative, require a good sample preparation and purification, and only extensive chemical modifications can be detected [290]. The technique can however be useful in localizing the site(s) of alterations damage in certain regions of a protein if the epitope recognized is known [280]. Di-Tyr is produced through a variety of reactions in which oxygen, metals and light can be involved. Therefore, appropriate sample handling is important during sample preparation.

The Tyr phenoxyl radical can form other oxidation products, such as iminoxyl radicals and 3,4-Dihydroxyphenylalanine (DOPA) [309]. The majority of these compounds are stable products, except DOPA, which can be oxidized further to a quinone compound and cyclized products (oxidation is pH dependent) [280]. These reactions may result in further radical formation [310]. DOPA itself is redox- active and has a sufficient reactivity to metals [311], Whereby in further reaction mechanism the production of superoxide anion is possible [312]. DOPA derivates can be identified after separation of the protein by SDS-PAGE via a redox staining method [304]. This method allows the detection of DOPA on an intact protein but is only qualitative due to the nonspecific reaction of DOPA with the staining reagent [280]. Aromatic residues, such as DOPA, o-Tyr, m-Tyr can be quantified with fluorescence spectroscopy (e.g., DOPA λ_max_ ≈ 280 nm [312]) [3,256], but the difference of Trp in comparison to these residues is virtually indistinguishable under normal conditions [312]. Peptide bound DOPA can be identified with (catechol)borate UV difference spectroscopy [305], but the method suffers from numerous interferences including those from Tyr and Trp [312]. Accurate determination of especially very low DOPA concentrations can be achieved with HPLC, combined with fluorescence detection (λ_ex_ ≈ 280nm, λ_em_ ≈ 320 nm) [306]. For better sensitivity, DOPA can be derivatised with ethylendiamine [306]. Since acid hydrolysis steps may cause artificial protein oxidation, cloud-point extraction can be used to increase the sensitivity [312]. After cloud-point extraction, DOPA was identified by using RP-HPLC approaches, combined with a fluorescence detector. Besides, electrochemical detection plays a role in detection of oxidized Tyr oxidation products [307,313]. Kumarathsan et al. (2003) developed a HPLC-CoulArray and a HPLC-amperometry procedure, in which DOPA and m-, o- and p- Tyr can be simultaneously analyzed [308].

#### 4.2.3. Phenylalanine

*Phenylalanine* (Phe) (Figure 3 and Table 7) can absorb light in the UV range of the electromagnetic spectrum. Phe photolysis leads to triplet state formation (^3^Phe), followed by either cleavage to yield a benzyl radical cation or the formation of a phenyl radical cation and an electron [3,314]. Phe can be oxidized to form 2-hydroxytyrosine (o-Tyr) and 3-hydroxytyrosine (m-Tyr) [260] by several oxidants including ·OH, ROO· and RO·. Since these species are similar to Tyr oxidation products [315], they can be detected with the analytical methods described above [316,317,318].

#### 4.2.4. Methionine

Oxidation of *methionine* (Met) (Figure 4 and Table 8) residues in therapeutic proteins are a common posttranslational modification, which can affect the bioactivity of the antibody and product quality of the antibody [319,320,321]. In a first oxidation step, Met oxidizes to the more polar Met sulfoxide [321] and in a second less common step to Met sulfone [322]. Met can undergo oxidation through a double electron transfer such as peroxides [323] or a single electron transfer such as photo-irradiation [324] (e.g., ^1^O_2_ ^2^). An electrophoretic assay was developed to determine methionine sulfoxides in proteins [325]. The assay is based on modifications of methionine with methyl methansulfonate, resulting in positively charged sulfonium derivate, which is used to separate the unmodified from the modified species in electrophoresis [325]. Separation can be hindered by cysteine oxidation. Since cysteine is oxidized to sulfonic acid in the same protein, this negative charge can annul the positive charge from Met alkylation [325]. Investigating the mAb structure in more detail, oxidation may occur at two solvent exposed methionine residues located in the fragment crystallizable fragment region (Fc) next to C_H_2- C_H_3-interface (Met 252 and Met 428) [320,326]. Met oxidation in this region can lead to small conformational changes in CH2, CH3, and/ or the CH2-CH3 interface. These conformational changes can be identified by Protein A affinity chromatography, since the Met oxidation may lead to a reduction of the Fc gamma receptor binding affinity [320]. The affinity Protein A chromatography utilizes this property, as Protein A binds close to the Met residues located in the Fc region. The binding to Protein A becomes weaker due to the disturbance of the hydrophobic interaction between the Fc and the Protein A binding. The distance between the sulfur atom in Met252 and atoms in Protein A such of those in Phe124 have a distance less than 4Å, so that Met oxidation to a sulfoxide is very likely to disturb the hydrophobic interaction. This method has limited options for the separation of oxidized protein variants [321,327]. As Balakrishnan et al. (2018) have shown, there are other methods available to detect Met oxidation, such Raman spectroscopy and FTIR [319]. A strong Raman vC-S band at 702 cm^−1^ was found to be an ideal Met oxidation marker due to the absence of overlapping protein or excipient bands in its immediate proximity [319]. Nevertheless, the measurable oxidation level is determined to be more than 5–10% [319], and the required protein amount is high [328]. The most common method to monitor Met oxidation levels relies on LC-MS based methods [319,329]. They are indispensable due to their high sensitivity, resolution, selectivity and specificity [319,327]. The localization of modification sites is obtained by generating proteolytic fragments at the domain or peptide level prior to LC/MS or by top down approaches [323]. However, the method is time-consuming due to digestion steps lasts until hours and prone to artifacts, caused by denaturation, disulfide reduction, alkylation and enzymatic digestion preparing steps [319]. In addition, electron capture dissociation (ECD) and collision-induced dissociation (CID), as two complementary fragmentation techniques can be used to detect and localize methionine sulfoxide in peptides, using Fourier transform ion cyclotron resonance MS [330]. Since the Fourier transform ion cyclotron resonance MS is a high-resolution technique, the approach can be used to detect small intact proteins up to ca. 12 kDa with high accuracy (<5 ppm) [331]. As an orthogonal approach to MS, an NMR technique can be used to identify Met sulfoxide from Met. NMR can distinguish photo-oxidized products even with the same mass difference. The caveats of this method are the higher protein requirement and the relatively long measuring time [230]. In addition, immunological detection can be applied for methionine sulfoxide [332,333]. Furthermore, in biopharmaceutical industry Met is used as an antioxidant, which is known to prevent residues in proteins from oxidation [334,335,336,337]. It is important to note that the average percentage of total Met residues in a protein is only about 2%, while the percentage of surface-exposed Met is even lower due to its hydrophobicity [332]. The surface exposed Met residues are more susceptible to oxidation and are therefore being attacked by most oxidizing agents [332]. Free Met is added to buffer solutions, which is sterically unobstructed and likely used as radical scavenger [337,338].

#### 4.2.5. Cysteine

*Cysteine* (Cys) (Figure 5 and Table 9) is able to absorb light in the UV range 250–300 nm [2]. Multiple cysteine oxidation products can be formed including cystine (disulfide), mixed disulfides, thiosulfinates and oxy acids, such as sulfenic, sulfinic and sulfonic acid [339]. The reaction between thiol and an alkene via the thiolene click approach to receive C-S linkages has been described under UV light [340]. In recent years, the reaction was also seen in the visible light spectrum with photo-catalysts [341,342,343]. Radical oxidants, such as alkyoxy radicals, superoxide radical anion and two-electron oxidants, are able to oxidize Cys [136]. One electron oxidants, such as superoxide are able to react with thiolates and generate a thiyl radical, which form with another thiolate a disulfide [179,344,345]. A few detection methods are based on the sulfenic acid chemical reactivity due to sulfur, which has an electrophilic and weak nucleophilic character. Thioles can easily be detected by the reaction of sulfenic acid with thioles to produce a disulfide. For example, by using a spectrophotometric assay, in which TNB (5-mercapto-2-nitrobenzoic acid) ionizes TNB^2−^, which has a characteristic absorption maximum at 412 nm [346]. However, since TNB^2−^ reacts to DTNB (also known as Ellman’s reagent) under aerobic conditions, long standing times should be avoided [347,348]. DNTB undergoes hydrolysis at higher pH and the absorption maximum of TNB is pH dependent [349]. The use of 4,4′-dithiopyridine (4-DP) can be utilized to quantify protein thiols as well [350]. 4-DP generates a strong absorbing resonance-stabilized tautomer, which has a pH independent absorption. Spectrophotometric methods are limited in their utility, since a low sensitivity requires a relatively large sample amount, and they are sensitive to contaminants such as thiol-containing small molecules [351]. Fluorescent labels, such as N-(1-pyrenyl)maleimide [352] and ThioGlo1 [353,354] show a better sensitivity.

In contrast to thiols, which can be directly detected due to their relatively high reactivity, disulfides need to be detected after reduction to the corresponding thiols. Indirect sulfhydryl detection methods require a protection of free thiols by using alkylating agents such as iodoacetamide, N-ethylmaleimide or iodoacetic acid [365,366]. Disulfides can be detected by using the same methods as for thiols. To quantify thiols and disulfides together, free thiol concentration must be quantified, followed by alkylation, reduction of disulfide bonds and subsequent quantification of additionally exposed thiols [367]. Reducing agents, such as dithiothreitol (DTT), 2-mercaptoethanol and substituted phosphines, such as tris(2-carboyethyl)phosphine are often used due to their quantitative and fast reaction without significant side reactions [355]. An example for this procedure can be provided for SDS-PAGE analysis. First free thiols are blocked and then disulfides reduced with DTT. Secondly, the new exposed thiols can then be labelled with 4-acetamido-4′-maleimidylstilbene-2,2disulfonate (AMS) [349]. approximately ca. 500 Da, which causes a detectable mobility shift when running SDS-PAGE [356]. Other thiol labeling methods can be adapted for disulfide analysis such as ^3^H-IAM [357], 5-iodoacetamidofluorescein [368] and biotin-conjugated iodoacetamide [369]. Direct quantification of disulfides is complex due to disulfide reduction and thiol-disulfide exchanges but may be achieved under mild protein digestion methods (e.g., using chemicals such as bromocyanine or enzymatic approaches) [370]. The reagents cleavage the polypeptide backbone between the half-cystinyl residues and minimize other reduction [370]. Another direct identification of disulfides can be achieved by “top down” MS workflows, which provides information to identity and locate small *m*/*z* modifications, such as disulfide bonds (disulfide bond results only in a 2Da reduction of molecular weight). “Top-down” MS can be useful since the technique avoids possible disulfide rearrangement which usually occurs in mild alkaline conditions during trypsin digestion. However, most LC-MS/MS coupled with database search engines are primarily designed for linear peptides only. Thus, so disulfide binding depends heavily on manual interpretation [358]. For further information about MS disulfide identification see Tsai et al. (2013) [358]. NMR approaches were investigated to detect disulfide bonds in proteins [359,371,372]. NMR approaches require a high concentration of large amount of homogenous proteins, and the approaches are limited to detect high-oxidized proteins. X-ray crystallographic studies provide information about the localization, but sample preparation is very time-consuming. For X-ray crystallography and NMR highly specialized skills and devoted efforts of structural biologists are demanded [359,360].

Commonly, sulfenic acids are formed in proteins through oxidation of the thiolate side chain of Cys by ROS such as H_2_O_2_ [349]. The selective reduction of sulfenic acid by arsenite can be used to detect sulfenic acid [365]. Free thiols are first blocked with maleimide [365,366] in the presence of SDS, followed by the reduction of sulfenic acids by sodium arsenite [349]. Some limitations accompany this technique, as a long-time scale accompanies the short-lived sulfonic acids. The assay is carried out under denaturing conditions, which could affect sulfonic acid residues and the selectivity of sodium arsenite is controversial discussed [349]. The reaction of sulfenic acid with thiols to produce disulfides has also been exploited to detect RSOH formation, as previously described. The detection of sulfenic acid residues has become increasingly divers due to a variety of chemical probes containing 1,3-diketone and its derivates. 1,3-diketone derivates can react directly with sulfenic acids. The most commonly used trap is dimedione (5,5-dimethyl-1,3-cycloheandione) and its derivatives. Since dimedione can only be detected directly through radiolabeling or MS, many derivates have been synthesized that contain fluorophores or affinity probes such as biotin tags [361,362,363]. These derivates can be detected by in-gel fluorescence and Western blot techniques. The sulfenic acid residues is a weak nucleophilic group, which enables the detection by the electrophilic reagent 7-chloro-4-nitrobenz-2-oxa-1,3-diazole [373]. Thus, the produced sulfoxide has a unique absorbance maximum. Indirect SOH-detection methods require protection of free thiols using alkylating agents as mentioned above (e.g., maleimide). Then reducing the protein-SOH by using arsenite and labeling nascent thiols with biotin-maleimide. Biotinylated reagents [374] (incorporated at the sites of sulfonic acid formation) enables the affinity isolation of labeled proteins. Following SDS-PAGE, in gel-digestion and MALDI-TOF MS analysis led to the identification of different proteins. Mass spectrometry is an excellent way to detect oxidation of sulfenic acid, which increases the mass of a protein by 16 Da. Direct analysis by mass spectrometry is limited by the inherent instability of the redox species under aerobic and/or denaturing conditions. Biotin switch assays offer a quantification method of sulfenic acid residues when combined with MS [375]. The application is limited due to the harsh conditions, which lead to fragmentation of biotin. Fragmentation of biotin can complicate MS analysis and database searching [376]. Thus, the biotin tag after the enrichment of the labeled protein should be removed [377,378,379,380]. Another technique represents antibodies, which are targeted against sulfenic acid residues in proteins [364]. These immunochemical approaches for the detection of sulfenic acid modification are highly specific and sensitive.

#### 4.2.6. Histidine

Therapeutic proteins are often buffered in *histidine* (His) (Figure 6 and Table 10) due to the stabilizing effect and its pI close to the desired pH stability optimum for mAbs [381,382]. Therefore, His oxidation can occur in the mAb sequence and in the buffered form, which can lead to intermolecular cross-links [383,384]. Oxidation of His mainly occurs via type II photo-oxidation mechanism. Thus, His is oxidized via ^1^O_2_ resulting in a cycloaddition, in which the imidazole ring reacts with ^1^O_2_ to form 2,4- or 2,5- EP intermediates [187,385,386,387]. Irradiation of free and protein-bound His yields a broad variety of different His oxidation products, including crosslinking products [9,187,383], such as His-His crosslinks [187] as well as ring opened products, such as Asp, Asn and 2-Oxo-His [388,389]. Stroop et al. (2011) reported about yellow coloring placebo formulations, after exposing them to light. They claimed about His-derived photo-sensitizers, which may initiate Trp photo-oxidation [390].

2-Oxo-His can be detected by RP-HPLC coupled with electrochemical detection (ECD), which offers a sensitive and specific detection method for oxidized proteins [391,392]. Protein cross-linked products as well as ring-opened derivates can be detected utilizing MS [187,388]. An oxidative modification of the side chain of His can significantly change the collision-induced dissociation (CID) of peptides containing oxidized residues [393]. The oxidation of His to 2-Oxo-His can lead to mismatches of oxidized residues in the CID spectra, when various isomers are simultaneously subjected to tandem MS [393,394]. However, mismatches can be avoided by using multiple stages of MS or by using electron-transfer dissociation (ETD) [393] to dissociate the oxidized peptides [9].

His residues have been shown to be involved in photo-oxidation induced cross-links. Various cross-links could be analyzed in mAbs such as His-His, His-Cys, His-Lys and His-Arg [187,221,255,395]. In Figure 6, only His-His is shown as representative example. Identification of these species can be achieved by using the ^18^O labelling MS approach as mentioned aforementioned.

#### 4.2.7. Lysine and Arginine

*Lysine* (Lys) (Figure 7 and Table 11 and Table 12) modifications in context of photo-oxidation are in most cases caused by radicals such as ·OH or RO·. They are generated via light induced electron transfer reactions. Lys can be hydroxylated to form hydroxylysine [370] or the epsilon amino group can be converted to an aldehyde to form α-aminoadipic semi-aldehyde [396]. Protein carbonyls such as the α-aminoadipic analysis can be identified with spectrophotometric analysis by using derivatization reagent, such as thiobarbituric acid (TBARS) or 2,4-dinitrophenylhydrazine (DNPH) [397,398]. These methods are useful in measuring the total levels of protein carbonyl but fail to provide information on the chemical nature of the modification. Akagawa et al. (2009) developed a technique using p-aminobenzoic acid (ABA) to derivatize the carbonyl and afterward quantify the amount with HPLC and fluorometric detection [399]. The ABA labeling was also used with LC-ESI-MS to determine α-aminoadipic in food proteins [400]. A limitation of current ABA-based methods is that they depend on complete acid hydrolysis of the protein and thus, information about the site of alteration is lost. Tandem LC-MS/MS)-based approaches enables the identification of protein carbonyls and the possible localization of carbonyl modifications in proteins at specific residues [401,402,403]. However, the low abundance of oxidative modified proteins requires a targeted approach for efficient tandem LC-MS/MS analysis. The use of a biotinylated hydroxylamine allows an affinity enrichment and thus, precise tandem MS/MS [404]. The oxidation product 5-Hydroxylysine can be detected via LC-MS/MS [405].

*Arginine* (Arg) (Figure 7 and Table 12) is susceptible to photo-oxidation. In addition, it is frequently used as stabilizing excipient in protein formulations [407]. Arg photo-oxidation was observed in alkaline solutions [408]. Arginine oxidation can lead to unstable hydroperoxides, 5-hydoxy-2-aminovaleric acid [409] and glutamic semi-aldehyde [260]. Since glutamic semi-aldehyde contains a carbonyl residue, the compound can be measured with the same analytic approaches described for α-aminoadipic semi-aldehyde. The oxidation residue 5-hydroxy-2-aminovaleric acid can be determined by using GC-MS [406].

## 5. Photo-Oxidation of Excipients

Biopharmaceuticals usually require the presence of additional cosolutes, such as buffer agents, antioxidants, sugars, or surfactants, which modulate the physicochemical characteristics of the entire drug product resulting in a greater resistance and stability against harmful environmental stress factors and improved manufacturing properties. In addition to biopharmaceuticals, these excipients are also potential targets of photo-oxidation processes, and some modifications are known to foster degradation processes [335,390].

Typical buffer systems for biopharmaceuticals are acetate, citrate, phosphate or histidine [382]. The latter was discussed in Section 4.2.6; Therefore, the main focus of this paragraph will be on the remaining buffering agents (Figure 8). In the case of acetate and phosphate buffered systems no adverse effects due to oxidation in context with protein stability have been currently described in the literature. In contrast, proteins can be modified by citric acid via a photo-chemical reaction at ambient light in the presence of trace levels of iron [410]. After light irradiation, the Fe^3+^ in the iron-citrate complex is reduced to Fe^2+^, and the citric acid molecule successively decomposes to several degradation products [411,412]. The intermediates which possess the potential for protein acetonation at the N-terminus or lysine residues can be determined by LC-MS and peptide mapping [412].

Since *polysorbates* (PS) (Figure 9) are used in pharmaceutical industry as surfactants of protein formulations, investigations on the photo-degradation are currently of high interest. Light exposure to PS solutions can result in structural changes of the PS molecule itself [413] via peroxide formation [191,414,415]. The formation of ROS species such as peroxides can result in oxidation of polysorbates [413]and of subsequent amino acid residues of proteins. However, the detailed mechanism and its consequences to proteins are not well understood yet [416]. Oxidation of polysorbate is complex to analyze due to its heterogeneity in nature of different (POE) units and fatty acid (FA) [417] (FA) composition. The unsaturated FA esters, as present in polysorbate 80 (PS80), are more susceptible to oxidation than their saturated pendants [417]. The increased susceptibility to oxidation with increasing amounts of double bonds can be explained due to carbon-centered radical stabilization. As consequence, oxidation leads to the formation of peroxides, aldehydes, ketones and epoxides [418]. In addition, oxidation can also occur at the POE chain, which can lead to POE-FA compounds [418]. The sorbitan compound can be oxidized as well, leading to glycol structures [419]. Intermediate radicals, which can arise from polysorbate light absorption, can oxidize the therapeutic protein [418,420]. Progress in development of analytical techniques has led to a number of polysorbate methods being based on RP-HPLC in combination with multiple detectors, e.g charged aerosol detector (CAD). As a result, a more robust detection of several different polysorbate components is possible. Polysorbate can also be measured by florescence micelle assay (FMA), which indirectly quantifies the increase in quantum yield of the fluorescent dye N-phenyl-1-naphathylamine (NPN) when incorporated into PS micelles. Therefore, the method is sensitive to intact PS, which is able to form micelles [421]. The NPN assay cannot directly distinguish degraded from intact PS. Thus, this micelle-based method would only detect changes affecting micelle formation, which could miss the loss of PS. NPN shows a different response to different species present in PS [422]. NPN has been shown to be more sensitive to PS polyester than to the main component of PS, the monoester [423]. NPN could possibly be more sensitive to monitor the oxidative loss of PSs, which shows more polyester degradation than chemical hydrolysis, which mainly affects the monoester fraction [423]. As discussed by Martos et al. (2017), NPN can interact with other hydrophobic components in the formulation, such as the protein itself and excipients. This increases the need for a strong NPN assay qualification to implement it for routine use in the pharmaceutical industry [423]. Polysorbate 20 (PS20) oxidation products can be detected by derivatization with DNPH (2,4-dinitrophenylhydazine) acting as oxidation marker [424]. Since few options are available to measure PS oxidation, quantification can be used to detect a decrease in polysorbate content. Thus, SEC has been described to quantify PS80 in formulations, but details on specification and precision are missing [425]. HPLC methods were developed with universal detectors, such as CAD [426] and evaporative light scattering detector (ELSD) to quantitate PS in protein formulations [427]. For further information see Kou et al. (2017) Fekete et al. (2010), Kranz et al. (2019), Li et al. (2014), Evers et al. (2020) [428,429,430,431]. To obtain a more precise determination of polysorbate derivatives, MS is often used [432,433,434,435]. Typically, LC-MS is used to analyze the nonvolatile and semi volatile fractions, whereas gas chromatography–mass spectrometry (GC-MS) is better suited for volatile molecules. Small molecules, such as aldehydes and acids, may arise from PS oxidation. The volatile oxidation degradation products of PS20, such as acetone, ethanol, pentane, hexane and heptanal, were detected [436]. Likewise, also for the oxidation products of PS80, which are acetone, ethanol, hexanal, heptanal, hexane, pentane, pentanal and butanal [436]. The degradants of other small molecules, such as aldehydes, ketones, alkanes, fatty ac-ids and fatty acid esters, can be detected by Stir-Bar-Assisted Sorptive Extraction GC-MS [437]. NMR spectroscopy is used to quantitate [438,439] and identify [423] major PS components, but without distinguishing structural variants. Since peroxides occur during the oxidative degradation of PS [415,440], the amount can be monitored as previously described for H_2_O_2_. Thus, it can be detected with the Amplex^®^ Red technique, which reacts with hydrogen peroxide to result in the formation of resorufin, which can be detected by fluorescence [441]. The LOD is 50 nM [423]. Furthermore, organic peroxides can be detected by a spectroscopic method named FOX assay [132]. Hydroperoxide mediate the oxidation of Fe^2+^ to Fe^3+^, which complexes with xylenol orange [125,415]. The complex is detected by absorbance at 560 nm with a LOD of 1 μM [423]. In addition, free radical formation can be determined by using NMR and EPR spin-trapping experiments. For further information see Section 3.3 and 3.4.

*Poloxamer* (Figure 9) is a polydisperse mixture of copolymers, consisting of hydrophobic poly(propylene oxide) (PPO) and hydrophilic poly(ethylene oxide) (PEO) blocks, which is used in biopharmaceutical formulations as surfactant [442]. So far, there are no published light degradation studies on poloxamer. However, an oxidative degradation after thermal exposure was observed. Costa et al. (1996) found small molecules, such as formaldehyde, acetaldehyde, and formic acid as degradation products [443]. The volatile products of degradation can be monitored by using GC-MS [444,445]. Only few detection options are described in literature to detect poloxamer oxidation. Quantification of poloxamer can be achieved via HPLC-SEC, offering a broad linear range and sufficient sensitivity, but the approach is not applicable in the presence of proteins or other complex formulations [446]. Sample preparation to remove the proteins are necessary to avoid column fouling. HPLC-RAM-ELSD method can be used for protein containing solutions, but the approach was only handled with low protein concentrations [447].

*Polyoxyethyleneglycolether* (PEG) (Figure 9), such as polyoxyethylene-23-laurylether (Brij-35), can be oxidized via light exposition [448]. Liu et al. (1999) published a quantitative approach of organic peroxides during the storage of Brij-solutions using HRP to form a fluorescent product [448]. This method is explained in more detail in the chapter ROS under peroxides (Section 3.3).

*Sugars or polyols* are often part of biopharmaceutical formulations as stabilizing agent for liquid and solid formulations [449,450,451,452]. The most common sugar or polyol molecules utilized in formulations are sucrose, trehalose, sorbitol, and mannitol [453], which are also known to act as radical scavengers and antioxidant [454,455,456]. Consequently, the addition of these stabilizers can partially negate the detrimental effects of photo-oxidation, that are based on ROS [457]. However, the disaccharides sucrose and trehalose bare the potential to release reducing monosaccharides, such as glucose and fructose upon hydrolysis, thereby promoting protein oxidation and the formation of advanced glycation end-products (AGEs) [458]. The latter are quite interesting concerning photo-oxidation processes, since AGEs can act as photo-sensitizers [459,460]. AGEs are initially formed via Maillard reactions between a reducing sugar such as glucose or fructose and a lysine or arginine residue [461] (Figure 10). These initial and simple AGEs can subsequently react with other molecules including ROS, thereby generating various different kinds of complex AGEs. Consequently, AGEs are a heterogeneous class of molecules, hence are investigated via different analytical techniques. A simple way for AGE detection are immunochemical techniques, such as ELISA or Western blotting. However, these techniques are limited to just a few common AGEs for which specific and expensive antibodies are available, such as against ε-N-carboxymethyllysine (CML) [462,463]. Nevertheless, the specificity and detection limit are useful in investigating certain AGEs down to 50 pmol per mg protein [464]. A good orthogonal and complementary approach in detecting and quantifying AGEs is the use of fluorescence spectroscopy. The excitation of AGEs, such as vesperlysine or pentosidine with UV light in the range of 300 to 400 nm, results in a strong fluorescent signal with emission maxima between 375 and 500 nm [465]. This approach is only capable to determine the total amount of fluorescent AGEs [466,467]. Therefore, to increase specificity and reduce potential interference by naturally fluorescent compounds such as NADH, fluorescence measurements are often coupled with HPLC analysis [468]. This approach allows the detection of multiple AGEs, such as pentosidine, argpyrimidine, or derivatized CML [467,469] with low detection limits like 20 fmol for pentosidine [470]. This approach can further be generalized by using MS as detection system to measure all AGEs beside fluorescent or derivatized species [471,472].

## 6. Conclusions

Degradation of therapeutic proteins due to photo-oxidation is a potential and always existing issue during manufacturing and storage procedures due to the direct exposure of the product to light. However, therapeutic protein formulations differ in their photo-degradation patterns, since these processes are influenced by several factors, that are often intertwined, such as the sequence and conformation of the therapeutic protein, the light exposure conditions (e.g., time, temperature, wavelength, intensity/flux), the presence of certain excipients and surfactants or solution pH. Hence, the investigation of these unique processes requires a target-oriented selection and combination of suitable analytical techniques. The analysis of e.g., light-induced radicals, poses challenges, because certain ROS are short-lived compounds demanding a more sophisticated approach, in which the compounds are either immediately detected or derivatized during sample irradiation. Therefore, acquiring a broad overview of established and novel techniques concerning photo-oxidation analysis is mandatory for investigating degradation pathways and detecting problematic or harmful byproducts. In addition, it will help to identify specific root causes for, e.g., the presence of certain impurities, that might act as a photo-sensitizer and thus induce photo-oxidation of the drug or an excipient. By understanding the mechanism of photo-degradation, it is possible to add certain excipients (antioxidants) to the formulation, such as Met, which may prevent photo-oxidation of the proteins.

In addition, it will help to identify specific root causes for e.g. the presence of certain impurities, that might act as a photo-sensitizer and thus induce photo-oxidation of the drug or an excipient. Understanding the mechanism of photo-degradation, it is possi-ble to add certain excipients (antioxidants) to the formulation, such as Met, which may prevent photo-oxidation of the proteins.

Consequently, future technical developments in this field will be beneficial not just for gaining further knowledge about light induced degradation processes, but also to further increase product safety and stability.

## Data Availability

Not applicable.

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
