# Peer review of "Photo-Oxidation of Therapeutic Protein Formulations: From Radical Formation to Analytical Techniques"

_pharmaceutics, 2021, doi:10.3390/pharmaceutics14010072_

Round 1
Reviewer 1 Report
General remark:
The paper needs to be subjected to editing of English language and style. Special attention has to be paid to repetitions, colloquialisms and imprecise statements. As an example:
‘In photometry, the radiant power at each wavelength is weighted by a luminosity function that models human brightness sensitivity. In photometry, the irradiance is called illuminance which is measured in lux (power per unit area; mW∙m-2) (…)’ (repetition)
‘The basis of light investigation is the measurement of light intensity, depending on the wavelengths of the light under investigation.’ (repetition)
‘it is generally a better reducing agent.’ (colloquialisms)
‘The first group of separation methods is roughly based on size such as size-exclusion chromatography (SEC),(…)’ (imprecise statement)
‘Worth mentioning, these wavelengths are usually absorbed by commercially available window glass or medicinal containers such as glass vials or polyethylene infusion bags.’ (imprecise statement)
‘(…) can be used to solve the problem.’ (colloquialisms)
‘Some reaction intermediates possess the potential for protein acetonation at the N-terminus or lysine residues that can be determined by LC-MS and peptide mapping’ (imprecise statement)
‘As photo-sensitizers they also can generate radicals, which are further capable of damaging proteins’(colloquialisms)
Moreover multiple times along the text the Authors refer to previous or next chapters in the following way:
‘The latter was already discussed in the previous chapter’
Although not an error per se, it negatively influences the readability of the manuscript.
Section-related remarks:
Introduction:
1) ‘various protein modifications and degradation products’
Protein modifications are often also degradation products
2) ‘Hence, most photo-oxidation processes observed in therapeutic protein formulations are not based on proteinogenic sensitizer such as Trp or Tyr, since these solely absorb light in the UV-C (200-280 nm, FUV) and UV-B (280-315 nm) regions.’
Although very important from the perspective of the manuscript, this statement is not followed up in section 4.
Measurement of Light Intensity:
1) The section should be reorganized to clearly separate fragments describing different measurement techniques and their limitations. Currently the section starts with actinometry, then describes radiometry, and then actinometry again.
2) ‘For UV measurements, the ICH guideline proposes a quinine actinometer, but its sensitivity to temperature pose application issues’
Authors should elaborate more on application issues of quinine actinometer since it is recommended in ICH Q1B Photostability Testing of New Active Substances and Medicinal Products. Moreover the Authors should also elaborate on this guidance in the Manuscript.
Photo-Oxidation and Oxygen:
In general, this section lacks the perspective of therapeutic protein formulations. It should be reorganized to clearly link studies on described ROS with current knowledge on protein formulation development. Additionally, it should familiarize reader with current approaches to study ROS formation in therapeutic protein formulations. As an example, the application of described probes should be discussed in terms of using them in the protein samples. In summary, reader via this chapter
Photo-Oxidation of Proteins via ROS and Detection of Specific Modifications
1) Since the one of the main topics of the manuscript are ‘Analytical Techniques’ which can be used to evaluate photo-oxidation of therapeutic proteins, those two chapters should be merged together, and divided into sub-sections describing each of the analytical techniques. The manuscript in current form informs the reader about the modifications of the amino acid residues and not the usability of each mentioned analytical method in the field. Additionally, each ‘analytical technique’ subsection should start with brief introduction to the technique, applications found in literature and end with its limitations.
2) ‘In IgG1 formulations between pH 5-7, fragmentation of the C-N- terminal restudies (Asp-Lys-Thr-His-ThrI in the heavy chain of the Fab domain and the upper hinge region of the Fc domains was observed by using H2O2.’ (and other similar)
In case of studies in which oxidation was obtained using different measure than light exposure (photo-oxidation), a strong justification of reference applicability to this manuscript should be added.
3) The second separation group is based on the chemistry and physicochemical properties of fragments, (…)’
Size is a physical property of the molecule
4) ‘High performance liquid chromatography (HPLC) methods,(…)’
A abbreviation HPLC was already used in the manuscript.
5) ‘Size exclusion chromatography (SEC), which based on the hydrodynamic size of proteins, is used for the analysis and quantification of soluble aggregates that are noncovalent and irreversible.’
SEC can as well be used to analyze the covalent aggregates. The Authors mention it even in the manuscript:
‘For oligomers containing intermolecular cross-links SEC, SDS-PAGE, SCX, charged-base fractional diagonal chromatography and multistep methods, can be used to isolate cross-linked from non-cross linked species.’
6) ‘It is still challenging to distinguish oxidized proteins from unoxidized ones’
After stating statement above, Authors refer to publication from 2009 (Ehrenshaft et al. (2009)). Since 2009 analytical techniques (mainly MS-based ones) have exhibited significant development in terms of distinguishing oxidized proteins from unoxidized ones.
7) ‘A linear correlation between the oxidation of these Met residues and a loss of neonatal Fc receptor was observed’
This sentence does not make sense.
8) ‘Since the Fourier transform ion cyclotron resonance MS is a high-resolution technique , the technique can be used to detect small intact proteins with modifications up to ca. 12 Da
This sentence does not make sense. Fourier transform ion cyclotron resonance MS is very accurate equipment (more accurate than 12 Da).
9) ‘Therapeutic proteins are often buffered in histidine (His) (Figure 6) due to the stabilizing effect and its pI close to the desired pH stability optimum for mAbs. Therefore, His oxidation can occur in the mAb sequence and in the buffered form, which can lead to intermolecular cross-links.’
‘Arginine is susceptible to photo-oxidation. In addition, the amino acid arginine (Arg) is frequently used as stabilizing excipient in protein solutions’
Authors have dedicated section (Photo-oxidation of Excipients) to cover this topic.
Photo-Oxidation of Excipients
1) Although section should describe the analytical techniques used to evaluate the effects of photo-oxidation on excipients in some cases it only presents excipient quantitation methods:
‘SEC-UV has been described to quantify PS80 in formulations, but details on specification and precision are missing’
‘Quantification of poloxamer can be achieved via HPLC-SEC, offering a wide linear range and sufficient sensitivity, but the approach is not applicable in the presence of proteins or other complex formulations’
In my opinion, the paper should not be accepted for publication in its current state in Pharmaceuticals.
Author Response
Please take a look at the attached file.

Reviewer 2 Report
I suggest the authors for minor changes and corrections state below.
- In the list of abbreviations, please include ATTA, AMS, BMPO, DBZTC, DMPO, DNPH, ELISA, FMA, FOX, FTIR, HDX, HMW, ICH, LC-MS, NADH, NPN and SOSG,
- It is highly recommended to provide summary table listing different ROS species, amino acids and their derivatives, corresponding analytical technics/assays and the most relevant references (as ‘author’ et.al and reference numbers). This will improve the readability and comprehension of this review.
- Typographical Corrections:
Manuscript Page # |
Original text with Typo in manuscript |
Correction Required |
3 |
Tetranethylpiperidine |
tetramethylpiperidine |
2 |
exactly at 253,7 nm |
253.7 nm |
5 |
to generate the specific red fluorescence |
fluorescent |
8 |
(For detailed species information see Davies et al. 2004 166). |
Remove left and right parentheses |
8 |
(see paragraph 5). |
See section 5 |
9 |
restudies (Asp-Lys-Thr-His-ThrI |
Residues (Asp-Lys-Thr-His-Thr) ? |
11 |
Tyr, Tyr-Trp, Tyr-Lis, |
Lys? Please verify |
13 |
The ionized di-Tyr chromophore, in which one of the two phenolic hydroxyl groups is dissociated, is responsible for the range emission of di-Tyr at 400 nm296. |
It is not clear what it means by range emission. Please correct the sentence for clarity. |
13 |
A variety of HPLC-based methods is currently used to identify |
are currently used to identify |
13 |
residues in proteins can be identity with |
Can be identified with |
13 |
RP-HPLC with fluores-cence 296-298 or electrochemical detection after proteolysis or acid digestion 296,299-301 |
This sentences seems in complete. Please provide full sentence. |
14 |
p- Tyr can be simultaneous analyzed |
Can be simultaneously |
15 |
In the last years, the reaction was also seen in the visible light spectrum with photo-catalysts |
In the recent years? Pease change the sentence to have a more clarity on what it means ‘the reaction was also seen’. |
17 |
when various isomers are simultaneous subjected to |
Simultaneously |
18 |
glutamic semi-aldehyd 261. Since glutamic semi-aldehyd |
Aldehyde |
19 |
For further information see chapter 2. |
Refer it appropriately |
19 |
Liu et al. (429) published |
Correct year to be mentioned here |
20 |
(e.g. time, temperature, wavelength), |
Add intensity/flux to be included |
Scheme 2 |
.OOH is labelled as superoxide anion |
Please provide correct name as hydroperoxyl radical or equivalent name |
- Grammatical and Punctuation Corrections:
Manuscript Page # |
Original Text |
Grammatical/punctuation correction Required |
10 |
Aggregate formation can occur for example through unfolding by frag-mentation or through side chain modification, for further information see Lévy et al |
Split this into two sentences as
Aggregate formation can occur for example through unfolding by frag-mentation or through side chain modification. For further information see Lévy et al |
10 |
techniques, such as analytical ultracentrifugation (AUC) can be used, for more information see below in paragraph |
Split this into two sentences as techniques, such as analytical ultracentrifugation (AUC) can be used. For more information see section#? |
11 |
The Hydrogen-Deuterium eXchange Mass Spectrometry (HDX-MS) using “bottom up” approach is applied to observe selec-tive conformational changes at Trp residues after selectively oxidizing them 249. Both gas chroma-tography mass spectrometry and LC-MS have been used to identify and quantify…… |
Use appropriate font in line with with rest of the document. |
12 |
” The Hydrogen-Deuterium eXchange Mass Spectrometry (HDX-MS) using “bottom up” approach is applied to …… …… 290 nm was suggested to be a Trp modification 278.” |
Use appropriate font in line with with rest of the document. |
20 |
may prevent photo-oxidation on the proteins |
photooxidation of the proteins |
Author Response

(The authors gave the same response as above.)

Round 2
Reviewer 1 Report
Introduction:
2) ‘Hence, most photo-oxidation processes observed in therapeutic protein formulations are not based on proteinogenic sensitizer such as Trp or Tyr, since these solely absorb light in the UV-C (200-280 nm, FUV) and UV-B (280-315 nm) regions.’
Although very important from the perspective of the manuscript, this statement is not followed up in section 4.
Thank you very much for this comment. In general, proteins are not directly damaged by visible light because they do not absorb light in this spectral range. In accordance with the literature, one possible explanation is the presence of a non-proteinogenic (extrinsic) photosensitizer. These compounds may be introduced as impurities from manufacturing steps or additives. Some of these molecules can absorb light in the visible range, producing reactive oxygen species. These reactive oxygen species can in turn damage other organic molecules, including proteins, and thus initiate the degradation process. Proteins are not damaged directly in comparison to UV light, but rather indirectly via reactive oxygen species when exposed to visible light. However, the basic photochemistry underlying both processes is similar. In both cases, the initial process of a proteinogenic and a non-proteinogenic photosensitizer proceeds via the type I and type II mechanism. The main difference is that the degradation process of non-proteinogenic photosensitizers requires the use of reactive oxygen species as a kind of mediator. Concerning realistic light conditions for biopharmaceuticals, observed photo-oxidation processes are in all likelihood based on non-proteinogenic photosensitizers. Therefore, we made this statement at the beginning of chapter 3 (chapters 3 and 4 have been merged) and referred to the review written by Schöneich, which explains impurities and other compounds in greater detail. Nevertheless, the intended focus of our manuscript was the detection of chemical modification of biopharmaceutical formulations induced via realistic (visible) light conditions.
Reviewers response (2nd review):
In chapter 3. Authors state:
'However, the basic principle underlying the UV and visible area mechanism is similar. In both types, the initial process starts with a proteinogenic or non-proteinogenic photosensitizer via type I and type II mechanism.'
The fragment suggest that visible area mechanism might act via proteinogenic photosensitizer which is in contrary to what Authors state in response presented above. Specifically:
'In accordance with the literature, one possible explanation is the presence of a non-proteinogenic (extrinsic) photosensitize'
This fragment should be clarified
Photo-Oxidation of Proteins via ROS and Detection of Specific Modifications
5) ‘Size exclusion chromatography (SEC), which based on the hydrodynamic size of proteins, is used for the analysis and quantification of soluble aggregates that are noncovalent and irreversible.’
SEC can as well be used to analyze the covalent aggregates. The Authors mention it even in the manuscript:
‘For oligomers containing intermolecular cross-links SEC, SDS-PAGE, SCX, charged-base fractional diagonal chromatography and multistep methods, can be used to isolate cross-linked from non-cross linked species.’
Thank you for your note, we specified between reducing and non-reducing SEC technique.
Reviewers response (2nd review):
Authors added sentence:
'Reducing SEC can also quantify covalent aggregates.'
Hypothetically, application of reducing conditions to SEC analysis would lower the total covalent aggregates content via the reduction of disulphide bonds. Authors should support the abovementioned sentence with a reference.
Photo-Oxidation of Excipients
1) Although section should describe the analytical techniques used to evaluate the effects of photo-oxidation on excipients in some cases it only presents excipient quantitation methods:
‘SEC-UV has been described to quantify PS80 in formulations, but details on specification and precision are missing’
‘Quantification of poloxamer can be achieved via HPLC-SEC, offering a wide linear range and sufficient sensitivity, but the approach is not applicable in the presence of proteins or other complex formulations’
In literature, only few methods are described to detect PS or poloxamer oxidation. In literature, the quantification poloxamer can be archived to detect a decrease in concentration upon oxidation. So it is an indirect method to detect poloxamer oxidation.
Reviewers response (2nd review):
Authors should consider also adding reference to the publication below:
Tingting Wang, Aaron Markham, Steven J. Thomas, Ning Wang, Lihua Huang, Matthew Clemens, Natarajan Rajagopalan, Solution Stability of Poloxamer 188 Under Stress Conditions, Journal of Pharmaceutical Sciences, Volume 108, Issue 3, 2019, Pages 1264-1271, ISSN 0022-3549,
https://doi.org/10.1016/j.xphs.2018.10.057.
Author Response
Please take a look at the attachment.
